# HCN2 channels in the ventral tegmental area regulate behavioral responses to chronic stress

Peng Zhong[1†], Casey R Vickstrom[1], Xiaojie Liu[1], Ying Hu[1], Laikang Yu[1], Han-Gang Yu[2], Qing-song Liu[1]*

[1]Department of Pharmacology and Toxicology, Medical College of Wisconsin, Milwaukee, United States; [2]Department of Physiology and Pharmacology, West Virginia University, Morgantown, United States

**Abstract** Dopamine neurons in the ventral tegmental area (VTA) are powerful regulators of depression-related behavior. Dopamine neuron activity is altered in chronic stress-based models of depression, but the underlying mechanisms remain incompletely understood. Here, we show that mice subject to chronic mild unpredictable stress (CMS) exhibit anxiety- and depressive-like behavior, which was associated with decreased VTA dopamine neuron firing in vivo and ex vivo. Dopamine neuron firing is governed by voltage-gated ion channels, in particular hyperpolarization-activated cyclic nucleotide-gated (HCN) channels. Following CMS, HCN-mediated currents were decreased in nucleus accumbens-projecting VTA dopamine neurons. Furthermore, shRNA-mediated HCN2 knockdown in the VTA was sufficient to recapitulate CMS-induced depressive- and anxiety-like behavior in stress-naïve mice, whereas VTA HCN2 overexpression largely prevented CMS-induced behavioral deficits. Together, these results reveal a critical role for HCN2 in regulating VTA dopamine neuronal activity and depressive-related behaviors.
DOI: https://doi.org/10.7554/eLife.32420.001

*For correspondence:
qsliu@mcw.edu

Present address: †Division of Neurobiology, Department of Molecular and Cell Biology, Helen Wills Neuroscience Institute, University of California, Berkeley, United States

Competing interests: The authors declare that no competing interests exist.

## Introduction

Depression is highly prevalent throughout the world population (*Kessler and Merikangas, 2004*). Clinically available antidepressants share the same core mechanisms of blocking serotonin and nor-adrenaline reuptake in the brain. However, a significant number of patients with depression do not fully respond to serotonin and/or noradrenaline reuptake inhibitors (*Al-Harbi, 2012*). There is an increasing appreciation for the role of dopamine in the pathophysiology of depression (*Nestler and Carlezon, 2006*). Levels of dopamine metabolites in cerebrospinal fluid are reduced in depressive subjects (*Bowden et al., 1997*). There is a high incidence (30–50%) of comorbid depression in patients with Parkinson's disease (*Burn, 2002*), which is characterized by the degeneration of midbrain dopaminergic neurons (*Alberico et al., 2015*). In patients with untreated Parkinson's disease, higher depression scores were associated with lower dopamine synthesis capacity in the striatum (*Joutsa et al., 2013*). Additionally, in a double-blind clinical study, the dopamine receptor agonist pramipexole produced antidepressant effects in patients that failed to respond to standard antidepressant treatments (*Franco-Chaves et al., 2013*). Given the well-established role of dopamine in reward processing and motivation (*Brischoux et al., 2009*; *Morales and Margolis, 2017*), and that anhedonia and lack of motivation are core symptoms of depression (*Duman, 2007*; *Nestler and Carlezon, 2006*), it is thought that dysfunction of the dopamine reward system might contribute to anhedonia and the loss of motivation common in depression (*Nestler and Carlezon, 2006*).

Chronic mild unpredictable stress (CMS) (*Willner, 2005*) and chronic social defeat stress (CSDS) (*Berton et al., 2006*; *Golden et al., 2011*; *Krishnan et al., 2007*; *Krishnan and Nestler, 2011*) are

well-established rodent models of depression. However, CMS and CSDS induce opposing changes in ventral tegmental area (VTA) dopamine neuron activity: CMS reduced action potential (AP) firing in VTA dopamine neurons (*Chang and Grace, 2014*; *Moreines et al., 2017*; *Tye et al., 2013*), whereas susceptibility to CSDS was accompanied by an increase in AP firing (*Cao et al., 2010*; *Chaudhury et al., 2013*; *Krishnan et al., 2007*; *Ku and Han, 2017*). The behavioral effects of optogenetically activating or inhibiting VTA dopamine neurons also differ between mice subject to CMS or CSDS. In the CSDS model, optogenetic phasic stimulation of VTA neurons that project to the nucleus accumbens (NAc) induced susceptibility to social defeat, whereas optogenetic inhibition of the VTA–NAc projection induced resilience (*Chaudhury et al., 2013*). In contrast, optogenetic phasic stimulation of VTA dopamine neurons reversed CMS-induced depressive-like behavior, while optogenetic inhibition of VTA dopamine neurons induced behavioral despair and decreased sucrose preference in stress-naïve mice (*Tye et al., 2013*). Thus, modulating VTA dopamine neuron firing can enhance or prevent the development of depression-like behavior, and potential factors that might explain these divergent results have been contemplated (*Hollon et al., 2015*; *Lammel et al., 2014*).

AP firing in VTA dopamine neurons is governed by multiple voltage-dependent ionic conductances, including the hyperpolarization-activated cation current ($I_h$), which is generated by hyperpolarization-activated cyclic nucleotide-gated (HCN) channels (*McDaid et al., 2008*; *Okamoto et al., 2006*). There are four subtypes of HCN (HCN1-4), which mainly form homotetramers with distinct properties (*Wahl-Schott and Biel, 2009*) and have different brain distributions (*Notomi and Shigemoto, 2004*). Although *Hcn1-4* mRNAs are expressed in the VTA (*Monteggia et al., 2000*), immunostaining for HCN protein has shown that HCN2 is the predominant HCN subunit expressed in the VTA (*Notomi and Shigemoto, 2004*). Tetratricopeptide repeat-containing Rab8b-interacting protein (TRIP8b) is an auxiliary subunit of HCN1-4 channels and is required for HCN trafficking and enrichment at dendrites (*Han et al., 2011*; *Lewis et al., 2009*; *Lewis et al., 2011*; *Santoro et al., 2009*). Global knockout of *Hcn1* or *Trip8b* reduced behavioral despair, but these mice did not display anxiolytic-like behaviors (*Lewis et al., 2011*), while lentivirus-mediated shRNA knockdown of HCN1 in the dorsal hippocampus promoted both anxiolytic- and antidepressant-like effects (*Kim et al., 2017*; *Kim et al., 2012*). The antidepressant-like effects of global *Trip8b* knockout can be reversed by reinstating TRIP8b expression in the dorsal hippocampus (*Han et al., 2017*; *Lyman et al., 2017*). Apathetic (*Hcn2^{ap/ap}*) mice with a spontaneous null mutation in *Hcn2* show reduced behavioral despair in the tail suspension test, but these mice exhibit deficits in motor coordination and locomotion and cannot perform the forced swim test due to an inability to swim (*Lewis et al., 2011*). Thus, HCN channels can be powerful regulators of depressive- and anxiety-like behaviors, and different HCN isoforms in different brain regions may play distinct roles in regulating rodent behavior.

The $I_h$ current generates pacemaker activity via permeability to cations (*Biel and Michalakis, 2009*), and enhancing $I_h$ current increases AP firing in VTA dopamine neurons (*Friedman et al., 2014*; *McDaid et al., 2008*; *Okamoto et al., 2006*). CSDS leads to an increase in $I_h$ current in VTA dopamine neurons in susceptible mice, and an even further increase in $I_h$ current in resilient mice, but normal AP firing is achieved in resilient mice via homeostatic upregulation of voltage-gated $K^+$ currents (*Cao et al., 2010*; *Friedman et al., 2014*). Overexpression of HCN2 in VTA dopamine neurons or intra-VTA infusion of an $I_h$ potentiator in susceptible animals produced a reversal of social avoidance behavior, likely by a homeostatic upregulation of $K^+$ current and the subsequent normalization of AP firing (*Friedman et al., 2014*). However, given that CMS and CSDS induce distinct neuroadaptations in VTA dopamine neurons (*Chaudhury et al., 2013*; *Hollon et al., 2015*; *Lammel et al., 2014*; *Tye et al., 2013*), in particular the observation that CMS leads to decreased VTA dopamine neuron firing, it is of interest to investigate whether CMS alters $I_h$ current in VTA dopamine neurons, and if so, whether the alteration of $I_h$ current contributes to the development of depressive-like behaviors.

VTA dopamine neurons are heterogeneous in their projection targets and HCN expression (*Gantz et al., 2017*; *Morales and Margolis, 2017*). In mice, neurons that project to the lateral shell of the NAc display large $I_h$ current, whereas those that project to the medial prefrontal cortex (mPFC), medial shell of the NAc, or basolateral amygdala exhibit minimal $I_h$ current (*Baimel et al., 2017*; *Lammel et al., 2008*; *Lammel et al., 2011*). Dopamine neurons projecting to the lateral shell of the NAc are thought to signal primary reward and salience (*Bromberg-Martin et al., 2010*; *Lammel et al., 2011*), and optogenetic stimulation of an excitatory input to dopamine neurons projecting to the NAc lateral shell elicits reward (*Lammel et al., 2012*). Thus, in the present study, we

investigated how CMS affects $I_h$ current in VTA dopamine neurons that project to the lateral shell of the NAc. Furthermore, we determined whether virus-mediated knockdown or overexpression of HCN2 in the VTA affects depression- and anxiety-like behaviors. We provide evidence that CMS led to a decrease in $I_h$ current in NAc lateral shell-projecting VTA dopamine neurons, and that manipulating HCN2 expression in the VTA can powerfully regulate depressive- and anxiety-like behavior.

## Results

### CMS induced depressive and anxiety-like behavior

About equal numbers of C57BL/6J and DAT-tdTomato reporter mice (generated by crossing $Slc6a3^{Cre+/-}$ mice with Ai9 mice; see Materials and methods) were exposed to a variety of mild stressors in an unpredictable manner for 5 weeks (CMS group), while age-matched, stress-naïve C57BL/6J and DAT-tdTomato littermates served as corresponding control groups, and did not receive any special treatments except normal handling and behavioral tests. We have previously shown that mice exposed to CMS for 5 weeks exhibit depressive-like behaviors (*Wang et al., 2010*; *Zhong et al., 2014a*; *Zhong et al., 2014b*), which can be reversed by chronic treatments with the antidepressant fluoxetine (*Wang et al., 2010*). C57BL/6J mice and DAT-tdTomato mice are on the same genetic background, and because there were no significant differences between these groups in the various behavioral tests (*Figure 1—figure supplement 1*, *Supplementary file 1*), the results were pooled (*Figure 1*). C57BL/6J mice were later used for in vivo electrophysiology and DAT-tdTomato mice were later used for ex vivo slice electrophysiology because the latter allowed unambiguous identification of VTA dopamine neurons (see below). The timeline of CMS, behavioral tests, and electrophysiology is listed in *Figure 1A*.

Consistent with our previous study (*Wang et al., 2010*), we found that body weight of the CMS group was significantly decreased compared with control mice ($t_{24} = 3.2$, p=0.004; *Figure 1B*). We then examined whether CMS leads to depressive- and anxiety-like behavior. First, to assess whether CMS-exposed mice exhibit anxiety-related behavior or altered locomotor activity, we used the open field test (OFT). Mice tend to avoid open spaces, and reduced time spent in the center of an open field has been correlated with anxiety- and depression-like behaviors in rodents (*El Yacoubi et al., 2003*). There was no significant difference between control and CMS groups in the total distance travelled ($t_{24} = 0.5$, p=0.615; *Figure 1C*); in contrast, the time spent in the center square of the open field was significantly decreased in the CMS group ($t_{24} = 2.9$, p=0.007; *Figure 1C*). Second, the sucrose preference test (SPT) was performed to assess anhedonia, a core symptom of depression (*Duman, 2007*). CMS significantly decreased sucrose preference compared with control mice ($t_{24} = 4.6$, p<0.001; *Figure 1D*). Third, the elevated plus maze (EPM) test was carried out. A decrease in entries into and/or time spent in the open arms indicates anxiety-like behavior (*Komada et al., 2008*; *Rodgers and Dalvi, 1997*). CMS did not affect entries into the open arms ($t_{24} = 0.1$, p=0.919; *Figure 1E*), but significantly decreased the time spent in the open arms ($t_{24} = 2.2$, p=0.036; *Figure 1E*). Fourth, to assess depressive- and anxiety-like behavior, we used the novelty-suppressed feeding (NSF) test (*Santarelli et al., 2003*). An increase in the latency to feed on food in the center of a novel open field is indicative of anxiety- and depressive-like behavior. CMS significantly increased the latency to feed in the novel environment ($t_{24} = 2.9$, p=0.008; *Figure 1F*), but did not alter the latency to feed in the home cage ($t_{24} = 1.0$, p=0.345; *Figure 1F*). Thus, the CMS-induced increase in the latency to feed in the novel environment cannot be explained by possible changes in appetite. Finally, the forced swim test (FST) was carried out. An increase in the immobility time in the FST suggests behavioral despair and depression (*Porsolt et al., 1977*). CMS significantly increased immobility time in the FST ($t_{24} = 5.5$, p<0.001; *Figure 1G*). Taken together, these results indicate that CMS leads to depressive- and anxiety-like behaviors.

### CMS induced a decrease in AP firing in vivo in VTA dopamine neurons

We examined whether AP firing in VTA dopamine neurons in vivo was altered in CMS-exposed C57BL/6J mice. One day after the last behavioral test, CMS-exposed mice and time-matched control mice (C57BL/6J) were anesthetized with urethane and placed in a stereotaxic frame, and in vivo single-unit recordings were made in the VTA using stereotaxic coordinates (AP −2.9 to −3.3 mm, ML 0.6 to 1.1 mm, DV −3.9 to −4.5 mm) (*Paxinos and Franklin, 2001*) (see Materials and Methods).

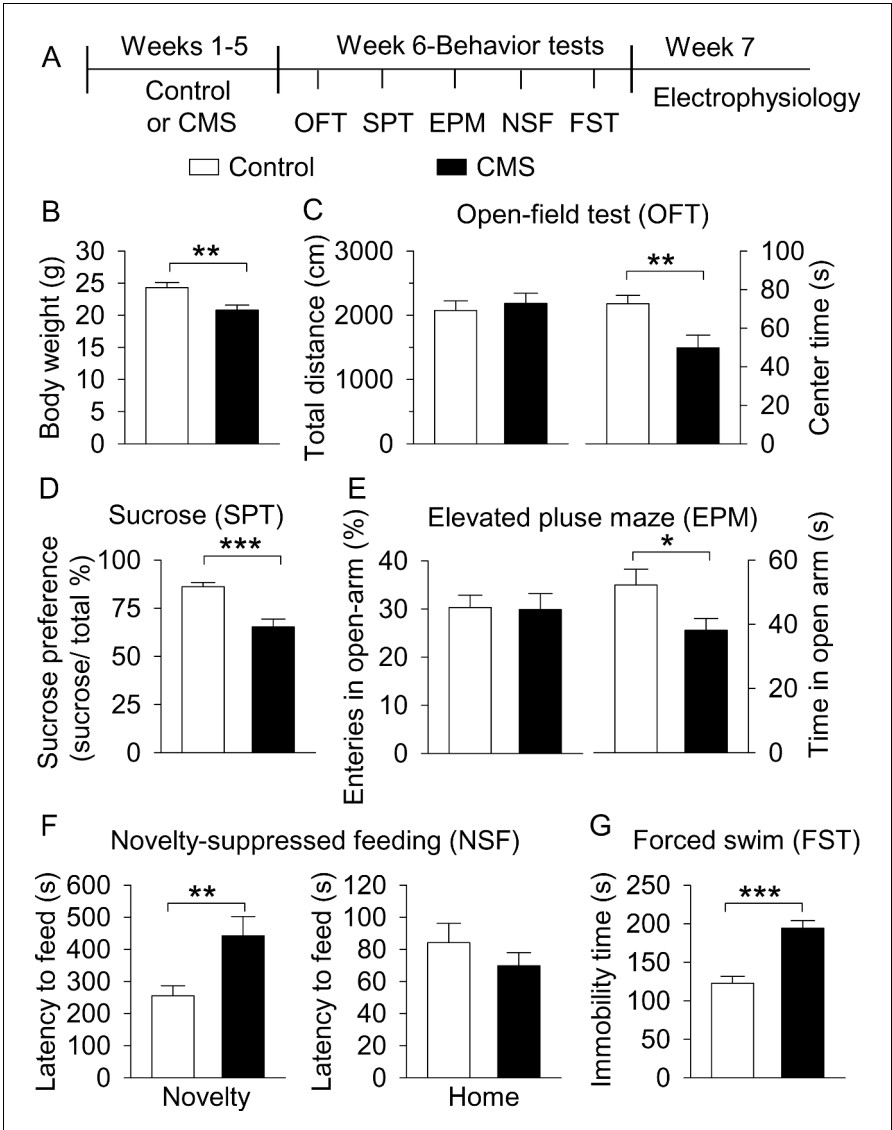

**Figure 1.** CMS produced depressive- and anxiety-like behaviors. (**A**) The timeline of CMS, behavioral tests, and electrophysiology. (**B**) CMS significantly decreased body weight compared with non-stressed control mice (**p=0.004, control, n = 14 mice; CMS n = 12 mice from **B** to **G**). (**C**) CMS significantly decreased the center time (**p=0.007) without affecting total distance traveled in the OFT (p=0.615). (**D**) CMS significantly decreased sucrose preference compared to control (***p<0.001). (**E**) CMS did not affect entries into the open arms (p=0.919), but significantly decreased the time spent in the open arms (*p=0.036) in the EPM test. (**F**) CMS significantly increased the latency to feed in the novel environment (Novelty) in the NSF test (**p=0.008) but did not significantly affect the latency to feed in the home cage (Home) (p=0.345). (**G**) CMS significantly increased immobility time in the FST (***p<0.001).

DOI: https://doi.org/10.7554/eLife.32420.002

The following source data and figure supplement are available for figure 1:

**Source data 1.** Body weight and behavior following CMS in *Figure 1B–G*.
DOI: https://doi.org/10.7554/eLife.32420.004

**Figure supplement 1.** C57BL/6J and DAT-tdTomato mice do not significantly differ in body weight or behaviors at baseline or in response to CMS.
DOI: https://doi.org/10.7554/eLife.32420.003

These coordinates targeted dopamine neurons in the lateral parabrachial pigmented nucleus (PBP), a subdivision of the VTA where dopamine neurons predominantly project to the lateral shell of the NAc, exhibit a large $I_h$ current (*Lammel et al., 2008*; *Lammel et al., 2011*), and are critically involved in reward and motivated behavior (*Lammel et al., 2011*; *Lammel et al., 2012*). Dopamine neurons were identified by a broad, triphasic AP of a width greater than 2 ms and a relatively slow firing rate (<10 Hz) (*Figure 2A*) (*Ungless and Grace, 2012*; *Ungless et al., 2004*). Dopamine neurons and recording locations were further validated post-mortem via juxtacellular labelling with neurobiotin and tyrosine hydroxylase (TH, a marker for dopamine neurons) immunostaining (*Chaudhury et al., 2013*; *Ungless et al., 2004*) (*Figure 2B*). Consistent with the electrophysiological criteria for identifying dopamine neurons (*Ungless and Grace, 2012*; *Ungless et al., 2004*), all neurobiotin-labeled neurons (control, n = 4; CMS, n = 5) we identified as dopamine neurons electrophysiologically were TH$^+$ (control, n = 4 mice; CMS, n = 5 mice). Thus, the waveform identification of the remaining putative dopamine neurons is likely sufficiently accurate. We found that the number of firing dopamine neurons encountered per electrode track ('population activity') was significantly decreased in CMS-exposed mice ($t_7$ = 7.3, p<0.001; *Figure 2C*), as well as the average AP firing rate in each neuron ($t_{30}$ = 2.1, p=0.045; *Figure 2A,D*). The proportion of spikes occurring within bursts was decreased in CMS-exposed mice ($t_{21}$ = 3.1, p=0.006; *Figure 2A,E*). Thus, CMS decreased the population activity,

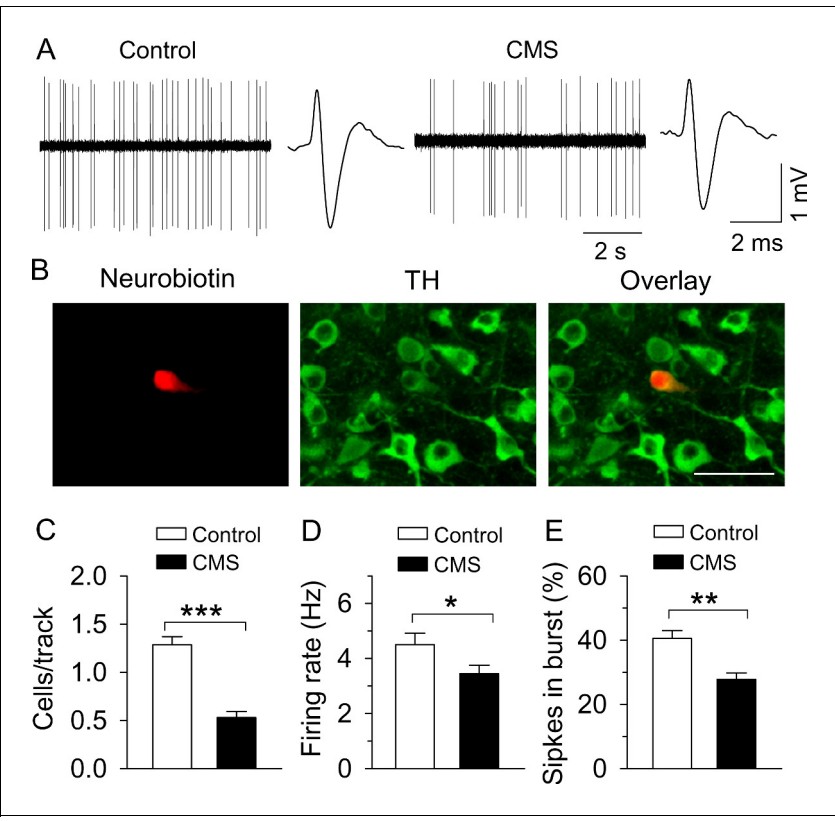

**Figure 2.** CMS decreased single-unit AP firing in VTA dopamine neurons in vivo. (**A**) Sample traces of VTA dopamine neuron AP firing in control and CMS mice. Dopamine neurons were identified by a broad triphasic extracellular action potential of a width greater than 2 ms and a relatively slow firing rate (<10 Hz). (**B**) A recorded dopamine neuron was confirmed by neurobiotin (red) and TH (tyrosine hydroxylase, green) co-localization. (**C–E**) Population activity (**C**, ***p<0.001, control, n = 4 mice; CMS n = 5 mice), firing rate (**D**) *p=0.045, control, n = 15 cells from four mice; CMS, n = 17 cells from five mice), and the percent of spikes in burst (**E**, **p=0.006, control, n = 10 cells from four mice; CMS, n = 13 cells from five mice) were decreased in CMS mice.

DOI: https://doi.org/10.7554/eLife.32420.005

The following source data is available for figure 2:

**Source data 1.** In vivo VTA dopamine neuron firing following CMS in *Figure 2C–E*.

DOI: https://doi.org/10.7554/eLife.32420.006

as well as the average AP firing rate and the proportion of spikes occurring in bursts. The CMS-induced decrease in the population activity implies that the relative number of spontaneously firing dopamine neurons is decreased.

## CMS decreased $I_h$ current and AP firing in VTA dopamine neurons ex vivo

The hyperpolarization-activated cation current ($I_h$), mediated by HCN channels (*Biel and Michalakis, 2009*), generates pacemaker activity and can drive AP firing in VTA dopamine neurons (*McDaid et al., 2008*; *Okamoto et al., 2006*). We examined whether alteration of $I_h$ current contributes to the CMS-induced decrease in AP firing. Although electrophysiological characteristics can reliably identify dopamine neurons in vivo (*Ungless and Grace, 2012*; *Ungless et al., 2004*), electrophysiological criteria such as the presence of $I_h$ current are likely not sufficient for the identification of VTA dopamine neurons in brain slices (*Margolis et al., 2010*; *Margolis et al., 2006*). To overcome this limitation, we bred *Slc6a3*$^{Cre+/-}$ (DAT-Cre) mice with Ai9 reporter mice, which express tdTomato in the presence of Cre (*Madisen et al., 2010*), to produce DAT-tdTomato reporter mice. Some TH-Cre driver lines are poorly specific for dopamine neurons in mice (*Lammel et al., 2015*), and thus were not used. Immunofluorescence staining indicated that tdTomato was overlapped with TH (*Figure 3A*), indicating that tdTomato-positive neurons are exclusively dopamine neurons.

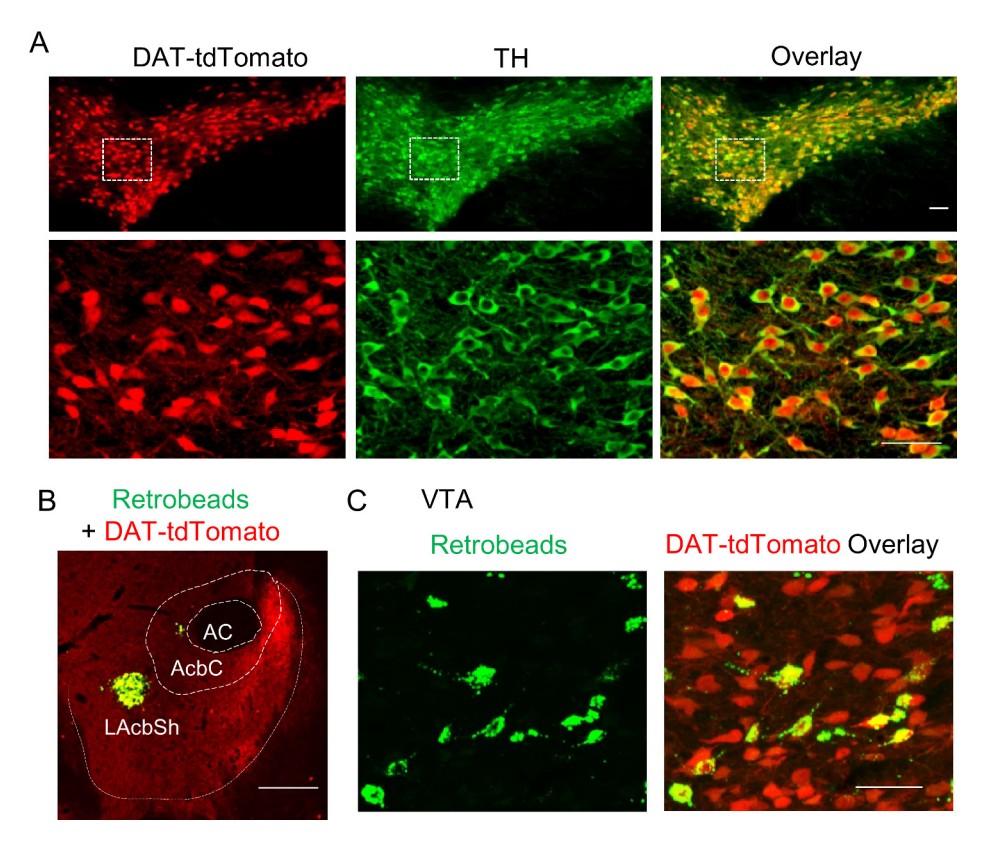

**Figure 3.** Retrobead labeling of VTA dopamine neurons that project to the lateral shell of the NAc. (**A**) DAT-Cre mice were bred with Ai9 reporter mice, which express tdTomato in the presence of Cre, to produce DAT-tdTomato mice. TdTomato and TH (green) were completely co-localized, indicating that tdTomato expression provides faithful reporting of dopamine neurons for slice physiology. (**B**) Green Retrobeads were injected into the lateral shell of the NAc (LAcbSh) in DAT-tdTomato mice. (**C**) The Retrobeads were retrogradely transported to the VTA and were predominantly co-localized with tdTomato-positive VTA dopamine neurons.
DOI: https://doi.org/10.7554/eLife.32420.007

VTA dopamine neurons that project to the lateral shell of the NAc exhibit a large $I_h$ current and play a primary role in reward encoding (*Baimel et al., 2017*; *Bromberg-Martin et al., 2010*; *Lammel et al., 2008*; *Lammel et al., 2011*; *Lammel et al., 2012*). Thus, we selectively performed slice electrophysiological recordings in these neurons. This was achieved by microinjecting green Retrobeads in the NAc lateral shell of DAT-tdTomato mice (*Figure 3B*) (see Materials and methods). Consistent with previous studies using TH staining (*Lammel et al., 2011*), we found that Retrobead-labeled neurons predominantly (95.7 ± 1.1%, n = 3 mice) were co-localized with tdTomato (*Figure 3C*), suggesting that the vast majority of VTA neurons projecting to the NAc lateral shell are dopamine neurons.

We investigated whether CMS altered $I_h$ current in NAc lateral shell-projecting VTA dopamine neurons from DAT-tdTomato reporter mice. $I_h$ current was recorded from neurons co-labeled with tdTomato (dopamine neurons) and green Retrobeads (neurons that project to the NAc lateral shell). $I_h$ current was induced by hyperpolarizing voltage steps (from −60 mV to −130 mV with −10 mV steps, 1.5 s duration) followed by a step to −130 mV for analysis of tail currents. $I_h$ amplitude, calculated by subtracting the instantaneous current ($I_{ins}$) from the steady-state current (*Figure 4A*), was significantly decreased in dopamine neurons from CMS-exposed mice at hyperpolarization steps to −90 mV or greater (amplitude: −90 mV, $t_{26}$ = 2.6, p=0.014; −100 mV, $t_{26}$ = 3.6, p=0.001; −110 mV, $t_{26}$ = 3.7, p<0.001; −120 mV, $t_{26}$ = 3.8, p<0.001; −130 mV, $t_{26}$ = 4.7, p<0.001; *Figure 4A,B*). This cannot be attributed to differences in cell size, since membrane capacitance ($C_m$) was not significantly different between control (50.9 ± 3.9 pF) and CMS mice (45.7 ± 2.2 pF, $t_{26}$ = 1.1, p=0.273; *Figure 4C*), and $I_h$ density, defined as $I_h$ amplitude at −130 mV normalized to cell capacitance, was significantly decreased in CMS mice compared with control mice ($t_{26}$ = 2.8, p=0.009; *Figure 4D*).

To assess potential differences in $I_h$ activation properties, tail current amplitudes were plotted as a function of test potentials and were fitted with a Boltzmann function to produce $I_h$ activation curves for both the control and CMS groups (*Figure 4A,E*). CMS significantly shifted the half-activation potential ($V_{1/2}$) to a more hyperpolarized potential ($t_{26}$ = 2.9, p=0.007; *Figure 4F*). Thus, CMS led to a significant reduction in $I_h$ current in NAc lateral shell-projecting VTA dopamine neurons, as well as a hyperpolarizing shift in the half-activation potential of $I_h$.

We measured the amplitude of $I_{ins}$ and plotted this current against the hyperpolarizing voltage steps. The slope of these I-V curves provides an approximation of the resting membrane conductance ($G_{resting}$) of the recorded neuron (*Baimel et al., 2017*; *Kimura et al., 1988*) (*Figure 4—figure supplement 1*). Although there was a trend toward a decrease in $G_{resting}$ in the CMS group compared with that of control, the difference did not reach statistical significance ($t_{26}$ = 1.7, p=0.107; *Figure 4G*). Thus, CMS leads to a decrease in $I_h$ current without significantly changing the resting membrane conductance.

Pharmacological manipulations that increase $I_h$ current lead to an increase in AP firing in VTA dopamine neurons (*Friedman et al., 2014*; *Wanat et al., 2008*), whereas the HCN blocker ZD7288 decreases AP firing in VTA dopamine neurons (*McDaid et al., 2008*; *Okamoto et al., 2006*). If CMS indeed leads to decreased $I_h$ current in dopamine neurons, and the CMS-induced decrease in $I_h$ current contributes to the decrease in AP firing observed in vivo, then an HCN blocker should produce a greater suppression of AP firing in control mice than CMS mice. Thus, we investigated the effects of the HCN channel inhibitor ZD7288 on $I_h$ current and VTA dopamine neuron AP firing. $I_h$ current was abolished by the HCN channel blocker ZD7288 (30 µM) in both the control group ($t_{10}$ = 7.6, p<0.001) and the CMS group ($t_{10}$ = 5.8, p<0.001; *Figure 5A*). We then determined whether ZD7288 altered AP firing rate in these two groups. Dopamine neurons ex vivo fire spontaneous, regular APs that lack bursting activity (*Johnson et al., 1992*; *McCutcheon et al., 2012*), which may be attributable to the severing of excitatory afferent inputs during slice cutting. Nevertheless, cell-attached AP recordings were made in NAc-projecting VTA dopamine neurons in the presence of CNQX (10 µM), AP-5 (20 µM) and picrotoxin (50 µM) to block excitatory and inhibitory synaptic transmission, thus helping to isolate cell-autonomous effects which may influence AP firing. Two-way ANOVA showed that CMS and ZD7288 had significant effects on the mean firing rate (CMS, $F_{1,47}$ = 13.7, p<0.001; ZD7288, $F_{1,47}$ = 28.6, p<0.001), with a significant interaction between CMS and ZD7288 ($F_{1,47}$ = 5.9, p=0.019; *Figure 5B,C*). Tukey's *post hoc* tests indicated that the mean firing rate was significantly decreased in the CMS group compared with the control group in vehicle-treated slices (p<0.001). Further, ZD7288 (30 µM) significantly decreased AP firing in both control (p<0.001) and CMS-exposed groups (p<0.05). The mean firing rate was not significantly different between control and

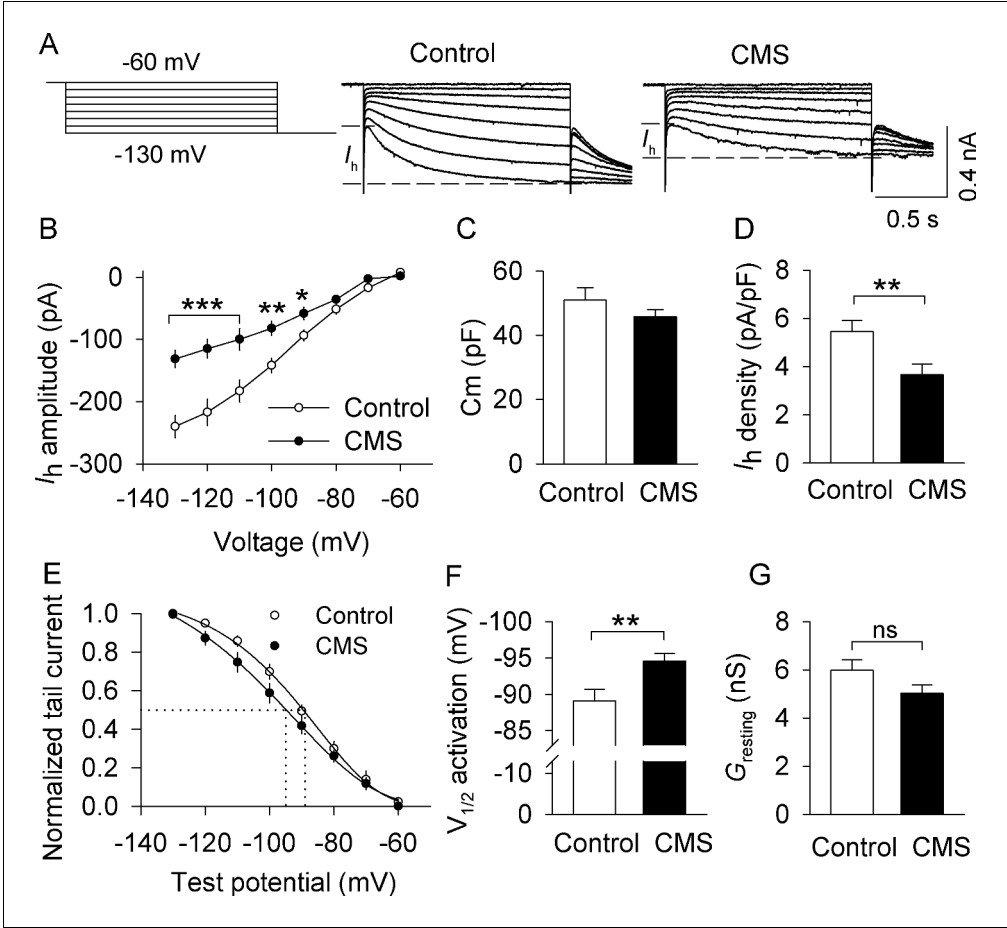

**Figure 4.** CMS decreased $I_h$ currents in VTA dopamine neurons that project to the lateral shell of the NAc (LAcbSh). (**A**) *Left*: Voltage protocol for recording $I_h$ current. *Right*: Representative $I_h$ current recorded from NAc-projecting VTA dopamine neurons in control and CMS mice. (**B**) Compared with the control group, $I_h$ amplitude was significantly decreased in the CMS group at corresponding hyperpolarization potentials (*p<0.05, **p<0.01, ***p<0.001, control, n = 15 cells from five mice; CMS, 13 cells from three mice from **B** to **G**). $I_h$ amplitude was calculated by subtracting the instantaneous current from the steady-state current achieved during the voltage step. (**C**) The membrane capacitance ($C_m$) was not significantly different between control and CMS mice (p=0.273). (**D**) $I_h$ current density was significantly decreased in the CMS group compared with the control group (**p=0.009). (**E**) $I_h$ activation curves in the control and CMS groups generated by the tail current protocol. Tail current amplitudes were fitted with a Boltzmann function. (**F**) CMS led to a significant hyperpolarizing shift of the half-activation potential ($V_{1/2}$) compared with that of control (**p=0.007). (**G**) The resting membrane conductance ($G_{resting}$) was not significantly different between control and CMS mice (p=0.107).
DOI: https://doi.org/10.7554/eLife.32420.008

The following source data and figure supplement are available for figure 4:

**Source data 1.** $I_h$ amplitude and activation properties in VTA dopamine neurons following CMS in *Figure 4B–G*.
DOI: https://doi.org/10.7554/eLife.32420.010
**Figure supplement 1.** The measurement of resting membrane conductance.
DOI: https://doi.org/10.7554/eLife.32420.009

CMS-exposed groups in the presence of ZD7288 (p=0.382; *Figure 5C*). The firing rate (FR) suppression (%) in the control group was significantly higher than that of the CMS-exposed group ($t_{21}$ = 3.2, p=0.004; *Figure 5D*). These results indicate that CMS exposure leads to decreased $I_h$ currents in NAc lateral shell-projecting VTA dopamine neurons, and that this likely contributes to the CMS-induced decrease in AP firing.

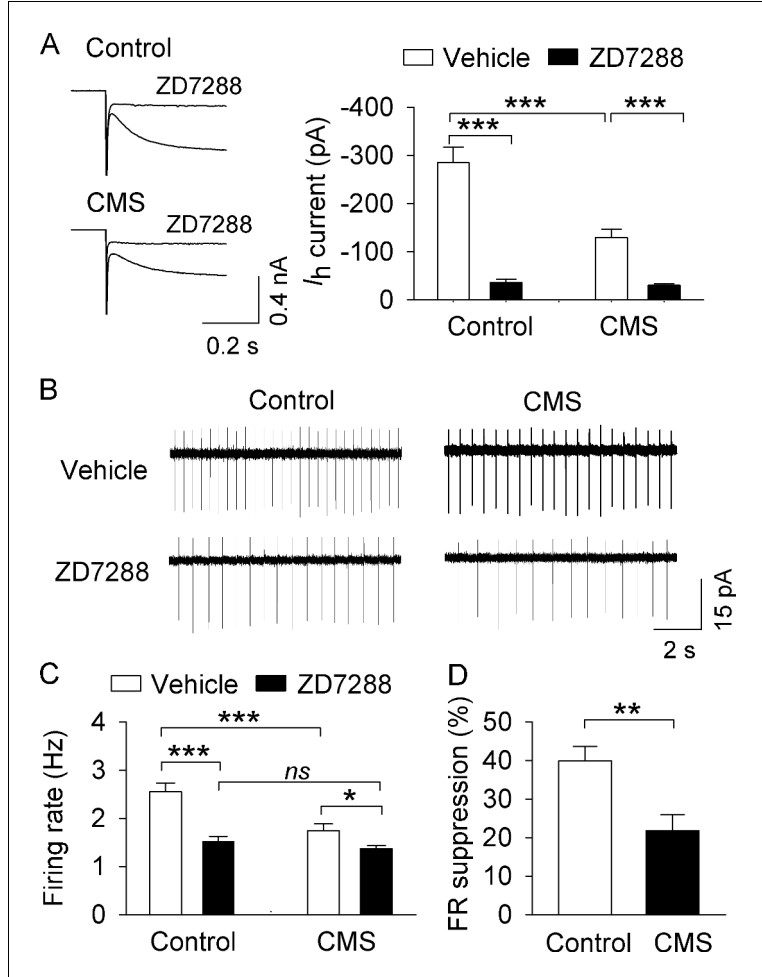

**Figure 5.** , CMS decreased AP firing in VTA dopamine neurons in midbrain slices. (**A**) $I_h$ currents recorded at −130 mV in NAc-projecting VTA dopamine neurons in both control (***$p<0.001$, n = 6 cells from three mice) and CMS groups (***$p<0.001$, n = 6 cells from three mice) were abolished by the $I_h$ channel blocker ZD7288 (30 µM). (**B**) Representative AP firing in cell-attached recordings from NAc-projecting VTA dopamine neurons in control and CMS slices before and after ZD7288 (30 µM). (**C**) The AP firing rate was significantly decreased in the CMS group (n = 13 cells from three mice) compared with the control group (n = 12 cells from four mice; ***$p<0.001$). ZD7288 significantly decreased the firing rate in both control (n = 11 cells from three mice) and CMS (n = 12 cells from five mice) groups (*$p<0.5$, ***$p<0.001$). The mean firing rate was not significantly different between control (n = 11 cells from three mice) and CMS (n = 12 cells from four mice) groups following ZD7288 ($p=0.382$). (**D**) The firing rate (FR) suppression (%) by ZD7288 in the control group (n = 11 cells from three mice) was significantly higher than that of the CMS group (n = 12 cells from five mice; **$p=0.004$).

DOI: https://doi.org/10.7554/eLife.32420.011

The following source data is available for figure 5:

**Source data 1.** Effects of ZD7288 on $I_h$ current and ex vivo AP firing in VTA dopamine neurons following CMS in *Figure 5A,C,D*.

DOI: https://doi.org/10.7554/eLife.32420.012

## shRNA knockdown of HCN2 in the VTA produced depressive- and anxiety-like behavior

We asked whether the CMS-induced decrease in $I_h$ current in VTA dopamine neurons contributes to depressive- and anxiety-like behavior. To mimic HCN current downregulation, we used an RNA interference technique to knock down HCN2 in the VTA, which is the predominant HCN isoform in the VTA as determined via immunostaining (*Notomi and Shigemoto, 2004*). AAV2-HCN2-shRNA-eGFP,

which expresses a short hairpin RNA (shRNA) targeting *Hcn2* mRNA, or AAV2-scramble-shRNA-eGFP was microinjected into the VTA bilaterally in C57BL/6J mice and DAT-tdTomato mice. Three weeks after the AAV injections, immunofluorescence staining from C57BL/6J mice was performed. AAV2-scramble-shRNA-eGFP was expressed in $86.1 \pm 5.1\%$ of $TH^+$ dopamine neurons in the VTA, while AAV2-HCN2-shRNA-eGFP was expressed in $83.9 \pm 6.3\%$ of $TH^+$ dopamine neurons, as shown by co-labeling of eGFP with TH in the VTA (**Figure 6A,B,C**).

To determine whether AAV2-HCN2-shRNA-eGFP was effective in knocking down HCN2 in the VTA, we made whole-cell recordings from eGFP and tdTomato co-expressing dopamine neurons in the lateral PBP. $I_h$ current was greatly attenuated in cells infected with AAV2-HCN2-shRNA-eGFP compared with cells infected with AAV2-scramble-shRNA-eGFP, as shown by a significant decrease in the maximal magnitude of $I_h$ current at $-130$ mV ($t_{27} = 8.3$, p<0.001; **Figure 6D**). The resting

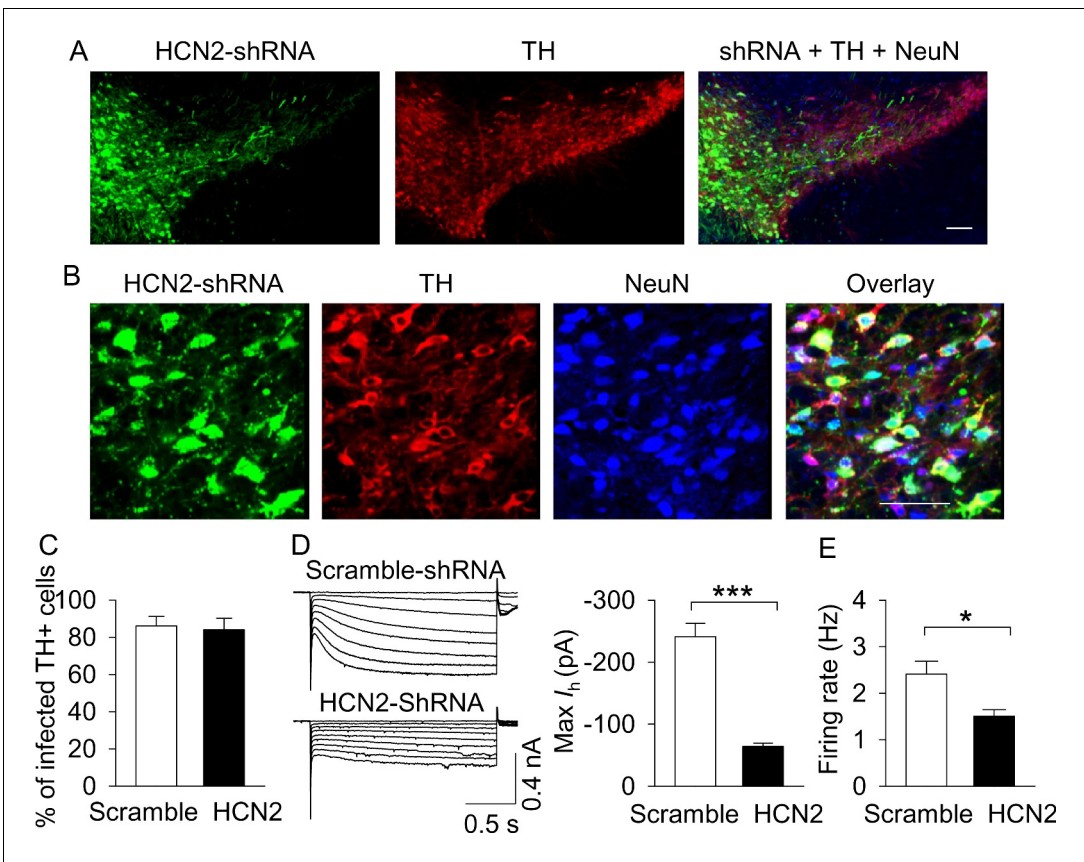

**Figure 6.** AAV-mediated shRNA knockdown of HCN2 in the VTA. (**A,B**), Immunofluorescence labeling showing the expression of AAV2-HCN2-shRNA-eGFP (green), TH (dopamine neuron marker, red) and NeuN (neuronal marker, blue) in the midbrain under low magnification (**A**) and high magnification (**B**). Scale bars: 50 µm. (**C**), The percentage of $TH^+$ VTA dopamine neurons that were infected with AAV2-HCN2-shRNA-eGFP or scramble-shRNA (n = 3 mice/group). (**D**), Maximal $I_h$ current amplitude was significantly decreased in AAV2-HCN2-shRNA-eGFP-infected dopamine neurons (n = 14 from three mice) compared with AAV2-scramble-shRNA-eGFP-infected dopamine neurons (n = 15 from four mice; ***p<0.001). (**E**), AP firing frequency was decreased in AAV2-HCN2-shRNA-GeFP-infected neurons (n = 10 from three mice) compared with AAV2-scramble-shRNA-eGFP-infected neurons (n = 9 from four mice; *p=0.036).

DOI: https://doi.org/10.7554/eLife.32420.013

The following source data and figure supplement are available for figure 6:

**Source data 1.** $I_h$ current and AP firing following shRNA-mediated HCN2 knockdown in **Figure 6C–E**.
DOI: https://doi.org/10.7554/eLife.32420.015

**Figure supplement 1.** The resting conductance ($G_{resting}$) was significantly decreased in the HCN-shRNA group (n = 14) compared with the scrambled group (n = 15, *p=0.022).
DOI: https://doi.org/10.7554/eLife.32420.014

conductance ($G_{resting}$) was significantly decreased following HCN2 knockdown ($t_{27}$ = 2.4, p=0.022; *Figure 6—figure supplement 1*), suggesting that $I_h$ current may also make a modest contribution to the instantaneous inward currents ($I_{ins}$). The AP firing rate of dopamine neurons in slices prepared from AAV2-HCN2-shRNA-eGFP-injected mice were also significantly decreased compared with that of AAV2-scramble-shRNA-eGFP-injected mice ($t_{17}$ = 2.3, p=0.036, *Figure 6E*).

Having confirmed the effectiveness of shRNA knockdown of HCN2 in the VTA, we next determined whether HCN2 in the VTA regulates depressive- and anxiety-like behaviors. Behavioral tests were conducted in C57BL/6J mice that received intra-VTA injection of AAV2-scramble-shRNA-eGFP or AAV2-HCN2-shRNA-eGFP three weeks prior. Compared with control (scramble-shRNA) mice, the

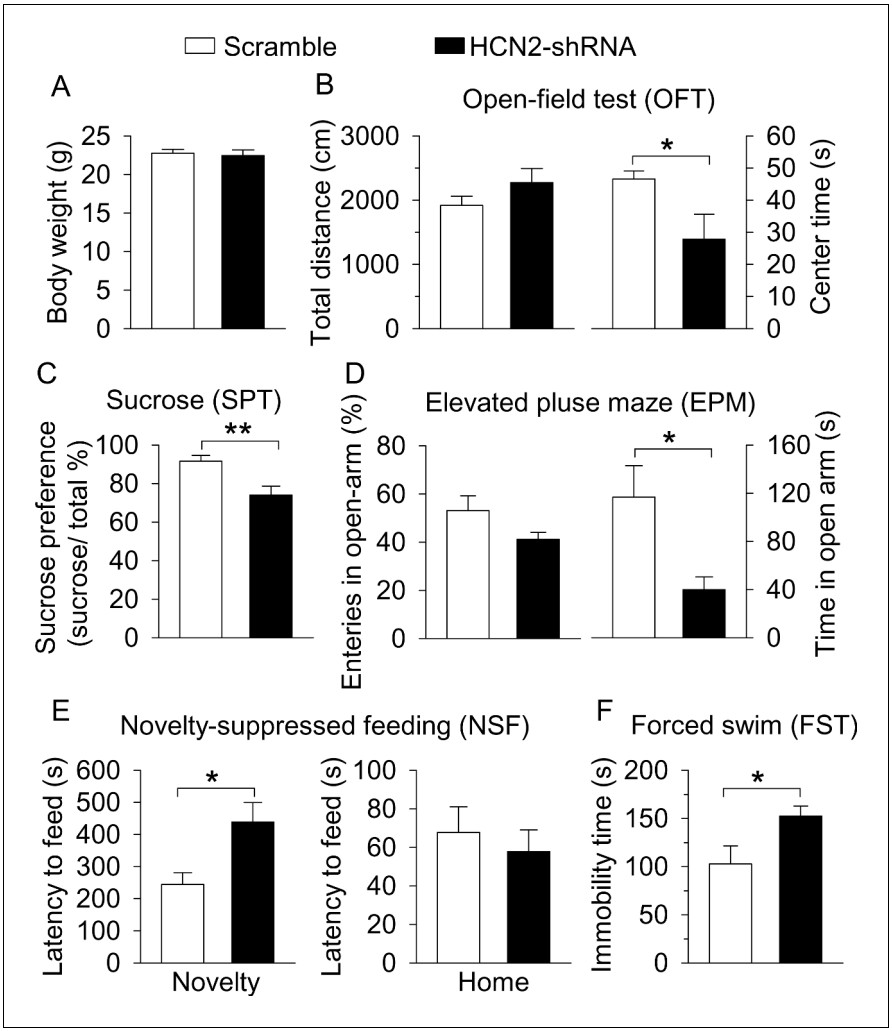

**Figure 7.** ShRNA knockdown of HCN2 in the VTA produced anxiety- and depressive-like behaviors. (**A**) VTA-specific HCN2 knockdown did not significantly affect the body weight of mice (control, n = 7 mice; CMS, n = 7 mice; p=0.727). (**B**) HCN2 knockdown significantly decreased the center time (*p=0.042) without affecting the total distance traveled (p=0.197) in the OFT. (**C**) HCN2 knockdown significantly decreased sucrose preference (**p=0.003). (**D**) HCN2 knockdown did not affect entries into the open arms (p=0.104) but significantly decreased time spent in the open arms (*p=0.018) in the EPM test. (**E**) HCN2 knockdown increased the latency to feed in the novel environment (Novelty) in the NSF test (*p=0.017) but did not significantly affect the latency to feed in the home cage (Home; p=0.583). (**F**) HCN2 knockdown increased immobility time in the FST (*p=0.039).
DOI: https://doi.org/10.7554/eLife.32420.016

The following source data is available for figure 7:

**Source data 1.** Body weight and behavior following VTA HCN2 knockdown in *Figure 7A–F*.
DOI: https://doi.org/10.7554/eLife.32420.017

body weight of the HCN2-shRNA group was not significantly different ($t_{12}$ = 0.4, p=0.727; *Figure 7A*). In the OFT, the total distance travelled was not significantly different ($t_{12}$ = 1.4, p=0.197; *Figure 7B*), but the time spent in the center square of the open field was significantly decreased in the HCN2-shRNA group ($t_{12}$ = 2.9, p=0.042; *Figure 7B*). The HCN2-shRNA group also showed a significant decrease in sucrose preference ($t_{12}$ = 3.7, p=0.003; *Figure 7C*). In the EPM test, there was no significant difference in the entries into the open arms ($t_{12}$ = 1.8, p=0.104; *Figure 7D*), but the time spent in the open arms was significantly decreased in the HCN2-shRNA group ($t_{12}$ = 2.7, p=0.018; *Figure 7D*). In the NSF test, the HCN2-shRNA group exhibited a significant increase in the latency to feed in the novel environment ($t_{12}$ = 2.8, p=0.017; *Figure 7E*), with no significant difference in the latency to feed in the home cage ($t_{12}$ = 0.6, p=0.583; *Figure 7E*). In the FST, the HCN2-shRNA group showed a significant increase in immobility time ($t_{12}$ = 2.3, p=0.039; *Figure 7F*). Taken together, these results indicate that shRNA knockdown of HCN2 in the VTA is sufficient to recapitulate the depressive- and anxiety-like behaviors seen following CMS exposure.

## Overexpression of HCN2 in the VTA prevented the development of CMS-induced depressive-like behavior

We have shown that CMS led to a decrease in $I_h$ current in VTA dopamine neurons, while shRNA knockdown of HCN2 recapitulated the depression- and anxiety-like behavioral effects of CMS. We next determined whether overexpression of HCN2 in the VTA could prevent the development of behavioral deficits produced by CMS. DAT-tdTomato mice received bilateral intra-VTA injections of AAV2-HCN2-eGFP or AAV2-eGFP. Three weeks after the AAV microinjections, midbrain slices were prepared and $I_h$ current was recorded. We found that $I_h$ current amplitude in HCN2-expressing dopamine neurons was significantly increased compared with eGFP-expressing dopamine neurons ($t_{27}$ = 3.3, p=0.002; *Figure 8A*). Thus, overexpression of HCN2 produces gain-of-function of $I_h$ current in the VTA.

Having demonstrated the effectiveness of HCN2 overexpression in enhancing $I_h$ current, we next examined the effects of VTA HCN2 overexpression on depression- and anxiety-related behaviors in control and CMS-exposed mice. AAV2-HCN2-eGFP or AAV2-eGFP was microinjected into the VTA bilaterally in C57BL/6J mice. After one week of recovery, mice were subjected to CMS or normal handling (non-stressed control) for 5 weeks, followed by behavioral testing.

Two-way ANOVA indicates that CMS and HCN2 overexpression had significant effects on body weight (CMS: $F_{1,36}$ = 11.1, p=0.002; HCN2: $F_{1,36}$ = 10.6, p=0.003; CMS x HCN2 interaction: $F_{1,36}$ = 4.7, p=0.038; *Figure 8B*). CMS-exposed mice showed a significant decrease in body weight compared with control mice in the AAV2-eGFP injection group (p<0.001), and this decrease was prevented by HCN2 overexpression (p<0.001; *Figure 8B*). Two-way ANOVA indicates that the total distance travelled in the OFT was not significantly different (CMS: $F_{1,36}$ = 0.01, p=0.977; HCN2: $F_{1,36}$ = 0.9, p=0.85; CMS x HCN2 interaction: $F_{1,36}$ = 0.4, p=0.549; *Figure 8C*). However, CMS and HCN2 overexpression had significant effects on the time spent in the center square of the open field (CMS: $F_{1,36}$ = 7.1, p=0.012; HCN2: $F_{1,36}$ = 5.6, p=0.023, CMS x HCN2 interaction, $F_{1,36}$ = 4.6, p=0.040; *Figure 8C*). Tukey's *post hoc* tests indicate that CMS significantly decreased the time spent in the center square in AAV2-eGFP-expressing mice compared with the stress-naïve group (p=0.002), but this decrease was prevented by HCN2 overexpression (p=0.004; *Figure 8C*). In the SPT, CMS and HCN2 overexpression had significant effects on sucrose preference (CMS: $F_{1,36}$ = 14.0, p<0.001; HCN2: $F_{1,36}$ = 21.7, p<0.001, CMS x HCN2 interaction: $F_{1,36}$ = 10.5, p=0.003; *Figure 8D*). Sucrose preference was significantly decreased in CMS mice compared to control mice (p<0.001) in the AAV2-eGFP injection group, but this decrease was prevented by HCN2 overexpression (p<0.001; *Figure 8D*). In the EPM test, neither CMS nor HCN2 overexpression had significant main effects on open arm entries (CMS: $F_{1,36}$ = 0.6, p=0.441; HCN2: $F_{1,36}$ = 0.3, p=0.604; CMS x HCN2 interaction: $F_{1,36}$ = 2.6, p=0.115; *Figure 8E*) nor the time spent in the open arms (CMS: $F_{1,36}$ = 0.7, p=0.413; HCN2: $F_{1,36}$ = 1.4, p=0.239; CMS x HCN2 interaction: $F_{1,36}$ = 2.1, p=0.159; *Figure 8E*). Planned comparison indicates that although CMS significantly decreased open arm time in AAV2-GFP-expressing mice (p=0.023), HCN overexpression did not significantly prevent the CMS-induced reduction in open arm time (p=0.057). In the NSF test, CMS and HCN2 overexpression had significant effects on the latency to feed in the novel environment (CMS: $F_{1,36}$ = 8.5, p=0.006; HCN2: $F_{1,36}$ = 8.3, p=0.007; CMS x HCN2 interaction: $F_{1,36}$ = 4.4, p=0.044; *Figure 8F*). Tukey's *post hoc* tests indicated that in the AAV2-eGFP injection group, CMS significantly increased the latency to feed

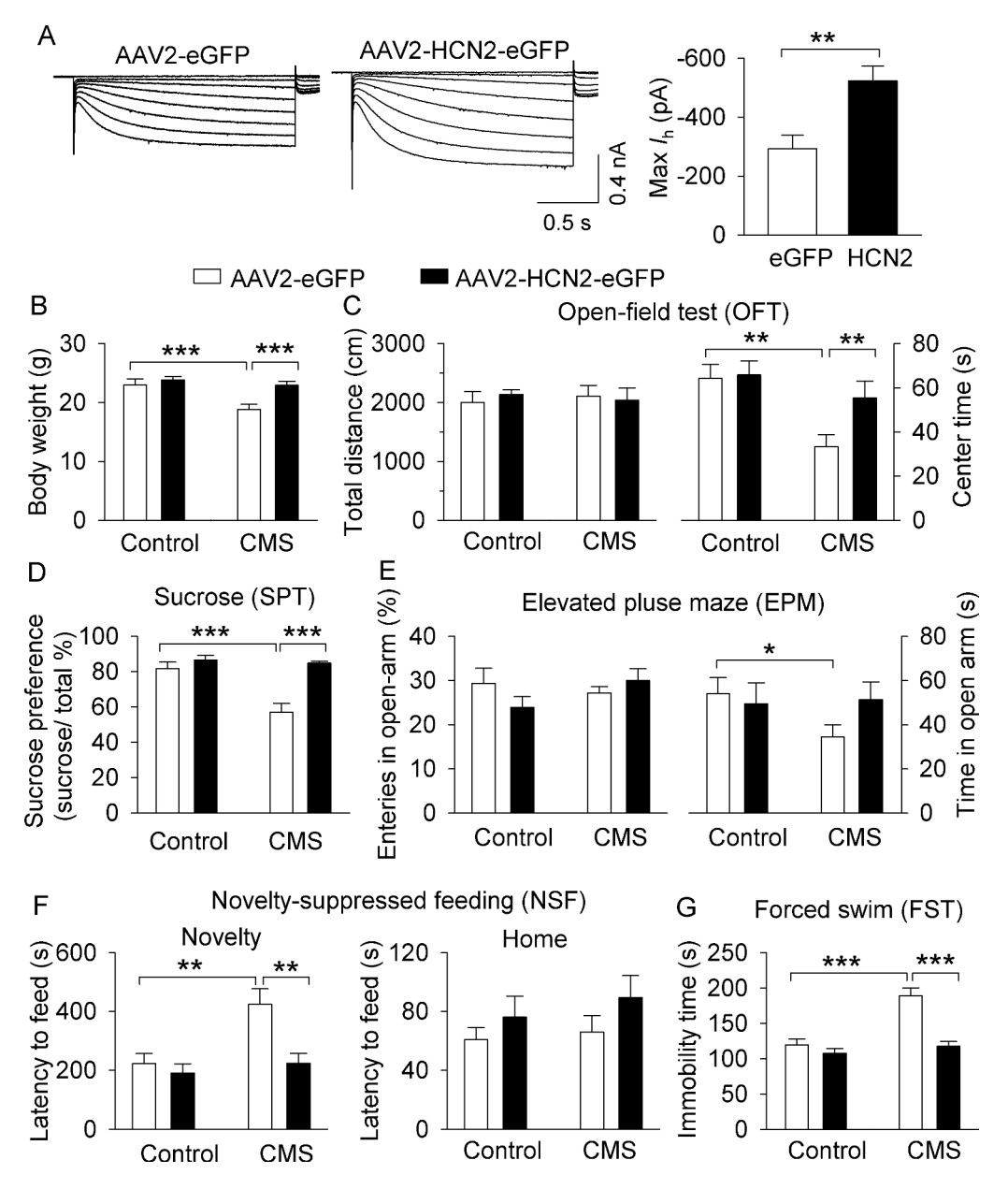

**Figure 8.** Overexpression of HCN2 in the VTA prevented the development of CMS-induced depressive-like behavior. (A) The maximal amplitude of $I_h$ current in AAV2-HCN2-eGFP-infected VTA dopamine neurons (n = 15 from four mice) was significantly increased compared with that in AAV2-eGFP-infected VTA dopamine neurons (n = 14 cells from five mice; **p=0.002). (B) CMS significantly decreased the body weight of mice in the AAV2-eGFP injection group (***p<0.001, control, n = 9 mice; CMS, n = 10 mice from B to G), whereas CMS did not decrease the body weight of mice in the HCN2 overexpression group (p=0.419, control, n = 10 mice; CMS, n = 8 mice from B to G). (C) Neither CMS nor HCN2 overexpression affected the total distance traveled in the OFT test (p>0.05). Compared with non-stressed control mice, CMS significantly decreased the time spent in the center square of the open field in the AAV2-eGFP group (**p=0.002), and this decrease was prevented by HCN2 overexpression (**p=0.004). (D) CMS significantly decreased sucrose preference in the AAV2-eGFP injection group (***p<0.001), whereas HCN2 overexpression prevented this decrease (***p<0.001). (E) In the AAV2-GFP group, CMS significantly decreased time spent in the open arms (*p=0.023) but did not affect open arm entries (p>0.05). HCN2 overexpression did not significantly affect open arm time nor entries (p=0.057) compared to the AAV2-GFP group. (F) CMS induced a significant increase in the latency to feed in the novel environment in the NSF test (**p=0.002), which was prevented by HCN2 overexpression (**p=0.001). Neither CMS nor HCN2 overexpression
*Figure 8 continued on next page*

*Figure 8 continued*

affected the latency to feed in the home cage (p>0.05). (**G**) CMS produced a significant increase in immobility in the FST (***p<0.001), and this increase was prevented by HCN2 overexpression (***p<0.001).

DOI: https://doi.org/10.7554/eLife.32420.018

The following source data is available for figure 8:

**Source data 1.** VTA HCN2 overexpression effects on $I_h$ current and CMS-induced changes in body weight and behavior in *Figure 8A–G*.

DOI: https://doi.org/10.7554/eLife.32420.019

(p=0.002). HCN2 overexpression prevented the increased latency to feed in CMS-exposed mice (p=0.001), but did not affect the latency to feed in control mice (p=0.573; *Figure 8F*). In contrast, neither CMS nor HCN2 overexpression affected the latency to feed in the home cage (CMS: $F_{1,36}$ = 0.6, p=0.463; HCN2: $F_{1,36}$ = 2.4, p=0.132; CMS x HCN2 interaction: $F_{1,36}$ = 0.1, p=0.748; *Figure 8F*), suggesting that changes in the latency to feed in the novel environment cannot be explained by changes in appetite. In the FST, CMS and HCN2 overexpression significantly affected the immobility time (CMS: $F_{1,36}$ = 20.6, p<0.001; HCN2: $F_{1,36}$ = 22.1, p<0.001; CMS x CHN2 interaction: $F_{1,36}$ = 11.4, p=0.002; *Figure 8G*). Tukey's *post hoc* tests indicated that in the AAV2-eGFP injection group, CMS significantly increased the immobility time (p<0.001). HCN2 overexpression prevented the CMS-induced increase in immobility time (p<0.001), but it did not affect immobility time in control mice (p=0.350; *Figure 8G*). These results indicate that HCN2 overexpression in the VTA prior to CMS exposure prevents the development of depressive-like behaviors.

## Discussion

VTA dopamine neuron AP firing is altered in chronic stress models of depression (*Cao et al., 2010*; *Chang and Grace, 2014*; *Moreines et al., 2017*; *Tye et al., 2013*), and modulating this firing can reverse or enhance the development and/or expression of depression-like behavior (*Chaudhury et al., 2013*; *Friedman et al., 2014*; *Tye et al., 2013*). Here, we have identified a putative mechanism linking CMS to decreased VTA dopamine neuron firing and the resulting behavioral deficits. We show in mice that following CMS, depressive- and anxiety-like behaviors and decreased VTA dopamine neuron firing are associated with reduced $I_h$ current and a hyperpolarizing shift in $I_h$ current in VTA dopamine neurons. Importantly, this decrease in $I_h$ current is behaviorally relevant, as shRNA knockdown of HCN2 in the VTA was sufficient to recapitulate depressive- and anxiety-like behavior, while VTA HCN2 overexpression prevented the CMS-induced development of depressive-like behavior. Thus, the downregulation of HCN2 channels in VTA dopamine neurons may be a primary contributor to the CMS-induced development of depressive- and anxiety-like behaviors.

CMS or other homotypic chronic stressors typically cause decreases in population activity, tonic and burst firing in VTA dopamine neurons (*Chang and Grace, 2014*; *Moore et al., 2001*; *Moreines et al., 2017*; *Tye et al., 2013*). We showed that the population activity, as well as the frequency of tonic and burst firing in VTA dopamine neurons were decreased in CMS-exposed mice. Importantly, our in vivo recordings targeted the lateral PBP, which is a more lateral subdivision of the VTA where dopamine neurons predominantly project to the lateral shell of the NAc (*Lammel et al., 2008*; *Lammel et al., 2011*) and play a primary role in reward and motivated behavior (*Lammel et al., 2011*; *Lammel et al., 2012*). Interestingly, CMS leads to decreased population activity in medial and central regions of the VTA but does not significantly alter population activity in the lateral VTA in the rat (*Moreines et al., 2017*). Further, no change in firing rate or burst firing was found in any VTA region (*Moreines et al., 2017*), consistent with their previous work (*Chang and Grace, 2014*; *Moore et al., 2001*). In contrast, our study and a previous study (*Tye et al., 2013*) found significant reductions in firing rate and burst firing following CMS, and we found decreased population activity in lateral VTA dopamine neurons. We suspect this discrepancy is due to differences in CMS paradigms and animal species, as the former studies applied 3–4 stressors/week over 4 weeks in rats (*Chang and Grace, 2014*; *Moreines et al., 2017*), whereas our study applied 14 stressors/week over 5 weeks in mice, and *Tye et al. (2013)* applied 14 stressors/week over 8–12 weeks in mice.

Both CMS and CSDS are commonly-used rodent models of depression (*Golden et al., 2011*; *Krishnan et al., 2007*; *Krishnan and Nestler, 2011*; *Willner, 2005*). Interestingly, CMS and CSDS produce distinct, even opposite effects on VTA dopamine neuron firing (*Chaudhury et al., 2013*; *Hollon et al., 2015*; *Tye et al., 2013*). CSDS, in contrast to CMS, increases tonic and burst firing in mice susceptible to CSDS, but not in resilient mice (*Anstrom et al., 2009*; *Cao et al., 2010*; *Chaudhury et al., 2013*; *Friedman et al., 2014*; *Krishnan et al., 2007*). Additionally, optogenetic phasic stimulation of NAc-projecting VTA neurons induced susceptibility to subthreshold social defeat stress, whereas optogenetic inhibition of the VTA–NAc projection induced resilience to CSDS (*Chaudhury et al., 2013*). However, optogenetic phasic stimulation of VTA dopamine neurons reversed the CMS-induced decrease in sucrose preference and increase in immobility in the tail suspension test (TST) (*Tye et al., 2013*). Interestingly, in non-stressed mice, optogenetic inhibition of VTA dopamine neurons was sufficient to reduce sucrose preference and increase immobility in the TST (*Tye et al., 2013*). Thus, these two models do not produce unitary effects on dopamine neuronal activity, and optogenetic manipulations of dopamine neuronal activity produce divergent behavioral changes, depending on the animal model used. Perhaps manipulations that *normalize* disruptions in AP firing induced by CMS or CSDS can produce antidepressant-like or resilient phenotypes.

In VTA dopamine neurons, $I_h$ current generates pacemaker activity together with other voltage-dependent ion channels. Pharmacological manipulations that increase $I_h$ current in VTA dopamine neurons lead to increased AP firing frequency (*Friedman et al., 2014*; *Wanat et al., 2008*), whereas the $I_h$ blocker ZD7288 reduces spontaneous AP firing in VTA dopamine neurons (*McDaid et al., 2008*; *Okamoto et al., 2006*). We therefore investigated whether CMS altered $I_h$ current in VTA dopamine neurons, as this may contribute to the CMS-induced reduction in AP firing. VTA dopamine neurons are heterogeneous in their afferent and efferent connectivity (*Brischoux et al., 2009*; *Morales and Margolis, 2017*) and electrophysiological properties (*Gantz et al., 2017*; *Lammel et al., 2008*; *Lammel et al., 2011*). Dopamine neurons in the lateral VTA typically fire with low-frequency (<10 Hz), project to the NAc lateral shell, and exhibit a large $I_h$ current, and both rewarding and aversive stimuli modify synapses onto these dopamine neurons (*Lammel et al., 2011*). CSDS increases $I_h$ current in NAc-projecting VTA neurons but not in mPFC-projecting VTA neurons in susceptible mice (*Cao et al., 2010*; *Friedman et al., 2014*). Interestingly, the amplitude of $I_h$ current in NAc-projecting VTA neurons is further increased in resilient mice, but normal AP firing in these mice is made possible via homeostatic upregulation of voltage-gated $K^+$ currents (*Friedman et al., 2014*; *Krishnan et al., 2007*). However, whether CMS affects $I_h$ current in these neurons had not been previously investigated.

Using Retrobead injections into the NAc lateral shell, we found that $I_h$ amplitude and $I_h$ current density were significantly decreased in NAc-projecting VTA dopamine neurons in CMS-exposed mice compared with that of control mice. CMS also led to a significant hyperpolarizing shift (~−5 mV) of $V_{1/2}$, which can reduce $I_h$ activation to hyperpolarizing stimuli. The mechanisms for this hyperpolarizing shift remain to be investigated. Additionally, we found that AP firing in VTA dopamine neurons ex vivo was significantly decreased in CMS-exposed mice. Unlike in vivo, the burst firing activity of VTA dopamine neurons is lacking ex vivo, likely due to the severing of synaptic inputs during slice cutting (*McDaid et al., 2008*). Nevertheless, we isolated the cell-autonomous effect of $I_h$ current on AP firing by performing cell-attached recordings in the presence of the glutamate receptor and $GABA_A$ receptor antagonists. We found that AP firing was decreased in CMS slices, and that the $I_h$ blocker ZD7288 produced a greater decrease in the firing rate in control slices than in CMS slices. Together, these studies indicate that CMS induced a decrease in $I_h$ current in VTA dopamine neurons, likely leading to the decrease in AP firing in vivo following CMS.

HCN channels contribute to resting membrane potential and conductance in neuronal cell bodies (*Banks et al., 1993*), dendrites (*Tsay et al., 2007*), and presynaptic terminals (*Huang and Trussell, 2011*). Consistent with this, we found that knockdown of HCN2 led to a significant decrease in resting membrane conductance, as derived from the instantaneous inward currents present immediately after hyperpolarizing voltage steps from a holding potential of −60 mV. Additionally, CMS led to a non-significant trend towards reduced resting conductance. These results together suggest that HCN2 channels modestly contribute to overall resting membrane conductance.

HCN1, HCN2 and the auxiliary subunit TRIP8b have previously been shown to regulate behavioral despair. *Hcn1* or *Trip8b* knockout leads to reduced immobility in the tail suspension test and forced

swim test, and $Hcn2^{ap/ap}$ mice show reduced immobility in the tail-suspension test, but these mice exhibit deficits in motor coordination and locomotion and cannot swim (*Lewis et al., 2011*). However, anxiety-like behavior is not altered in these three knockout mice in the elevated plus maze, dark/light box and marble burying tests (*Lewis et al., 2011*). Additionally, CMS induces an increase in HCN1 currents in the dorsal hippocampus, and shRNA knockdown of HCN1 in the dorsal hippocampus produced anxiolytic- and antidepressant-like behaviors (*Kim et al., 2017*, *2012*). Hippocampal CA1 pyramidal neurons usually do not fire spontaneous APs at resting membrane potential, and HCN1 channels at CA1 neuron distal dendrites limit neuronal excitability by reducing the integration of synaptic inputs (*Magee, 1998*, *1999*; *Tsay et al., 2007*), in part through interactions with M-type $K^+$ channels (*George et al., 2009*). Interestingly, depolarizing current injections elicited significantly more action potentials following HCN1 knockdown, knockout, and $I_h$ blockade in hippocampal neurons (*Kim et al., 2012*; *Lewis et al., 2011*). In contrast, consistent with previous studies (*McDaid et al., 2008*; *Okamoto et al., 2006*), we found that an HCN blocker or knockdown of HCN2 decreases AP firing in VTA dopamine neurons. Thus, it seems that the functional role of HCN channels depends on several factors, including the cellular location of HCN expression (e.g. distal dendrite vs. cell soma), the HCN isoform, the cell type in which HCN channels are expressed, and likely other factors. In VTA dopamine neurons, our results support a 'pro-excitability' role for HCN2, likely by generating a pacemaker current to support repetitive AP firing. It is possible that the differing neurophysiological functions of HCN channels in the dorsal hippocampus compared with VTA dopamine neurons may account for the opposing behavioral effects of HCN inhibition or activation in these structures.

Overexpression of HCN2 in VTA dopamine neurons, or intra-VTA infusion of the $I_h$ potentiator lamotrigine, produces the resilient phenotype in mice subjected to CSDS by activating a homeostatic mechanism leading to an upregulation of voltage-gated $K^+$ channels and normalization of AP firing (*Friedman et al., 2014*). Interestingly, intra-VTA infusion of an $I_h$ inhibitor can also promote the resilient phenotype following CSDS (*Cao et al., 2010*). How might an $I_h$ inhibitor lead to the same behavioral effect as $I_h$ potentiation? Whereas the $I_h$ inhibitor was applied acutely to CSDS-susceptible mice and presumably decreased abnormally elevated dopamine neuron firing (*Cao et al., 2010*), lamotrigine was administered repeatedly for five days and HCN2 was allowed to overexpress for four days before behavioral assessments were carried out (*Friedman et al., 2014*). $I_h$ potentiation led to an upregulation of $K^+$ currents in each scenario, leading to a reduction in otherwise abnormally elevated dopamine neuron excitability. In our study, CMS- or shRNA-mediated suppression of $I_h$ current leads to abnormally decreased dopamine neuron firing, and results in depressive- and anxiety-like behavior. Altogether, these findings support the notion that abnormal dopamine neuron firing is a unifying contributor to the development of depressive-like behavior.

Although shRNA-mediated HCN2 knockdown recapitulated the behavioral effects of CMS, it did not affect body weight. As body weight was measured 3 weeks after HCN2 knockdown, whereas the CMS-induced decrease in body weight was observed after 5 weeks of CMS, it is possible that body weight perturbations induced by HCN2 knockdown require additional time to manifest. Alternatively, HCN2 knockdown may not recapitulate all effects of CMS. Importantly, although the HCN2 knockdown was not cell type-specific, VTA dopamine neurons should be predominantly affected by the knockdown, as VTA GABA and glutamate neurons in mice exhibit little to no $I_h$ current (*Chieng et al., 2011*; *Hnasko et al., 2012*). Further, VTA dopamine neurons projecting to the mPFC, NAc medial shell, or basolateral amygdala exhibit minimal $I_h$ current (*Baimel et al., 2017*; *Lammel et al., 2008*; *Lammel et al., 2011*). Thus, NAc lateral shell-projecting VTA dopamine neurons should be the predominant VTA population affected by the knockdown.

Additionally, overexpression of HCN2 in the VTA prevented the majority of CMS-induced behavioral deficits and produced antidepressant-like effects. In the CSDS model, HCN2 overexpression selectively in NAc-projecting VTA neurons, but not in mPFC-projecting VTA neurons, normalizes social interaction deficits (*Friedman et al., 2014*). Thus, it is likely that enhancing $I_h$ current promotes the resilient phenotype in CSDS via effects on the VTA-NAc pathway. A limitation of our study is that although HCN2 overexpression is VTA-specific, it is not cell type- or projection-specific. As the VTA is heterogeneous with regard to cell type, connectivity, and behavioral function (*Morales and Margolis, 2017*), future studies employing more sophisticated viral tools for cell- and projection-specific overexpression of HCN2 will likely be necessary to parse out the contributions of individual VTA circuits to CMS-induced behavioral deficits. Nevertheless, a salient commonality between these

studies is that $I_h$ potentiation is capable of producing antidepressant-like phenotypes in both models, despite the fact that CSDS and CMS induced distinct neuroadaptations in VTA dopamine neuron activity and $I_h$ current.

HCN channels have been implicated in a variety of brain disorders, including epilepsy (*Benarroch, 2013*), Parkinson's disease (*Chan et al., 2011*), neuropathic pain (*DiFrancesco and DiFrancesco, 2015*), tinnitus (*Li et al., 2015*), and Neurofibromatosis type 1 (*Omrani et al., 2015*). Additionally, the rewarding effects of ethanol are believed to arise in part from activating HCN channels in VTA dopamine neurons (*Okamoto et al., 2006*; *Tateno and Robinson, 2011*). Consistent with this idea, overexpression of HCN2 in the VTA increases ethanol reward and consumption in rats (*Rivera-Meza et al., 2014*), perhaps through an increased ethanol-induced activation of $I_h$ current and AP firing in dopamine neurons. Further, chronic ethanol exposure can reduce $I_h$ current in VTA dopamine neurons (*Okamoto et al., 2006*). Given that alcohol use disorder (AUD) and major depressive disorder (MDD) are frequently comorbid in humans (*Yoon et al., 2015*), and that AUD can prospectively predict future MDD (*Boschloo et al., 2012*; *Brière et al., 2014*), a speculative mechanistic explanation is that chronic alcohol use leads to downregulation of HCN2 in VTA dopamine neurons, which may directly contribute to the development of depressive behavior. Given the widespread distribution (*Notomi and Shigemoto, 2004*) and functional diversity (*Santoro and Baram, 2003*) of HCN channels in the brain, it is likely that HCN channels play important roles in a wide variety of physiological and pathophysiological processes.

Clinical and animal studies suggest that VTA dopamine neurons and their major projection target, the NAc, play a critical role in the pathophysiology of depression. Depression and anxiety occur in 40–50% of patients with Parkinson's disease and these symptoms sometimes precede the appearance of motor dysfunction (*Burn, 2002*; *Dooneief et al., 1992*; *Taylor et al., 1986*). The dopamine agonist pramipexole produces antidepressant effects in patients who failed to respond to standard antidepressant treatments (*Franco-Chaves et al., 2013*). Different molecular changes in the VTA to NAc pathway confer vulnerability or resilience to chronic social defeat stress (CSDS) (*Berton et al., 2006*; *Francis and Lobo, 2017*; *Krishnan et al., 2007*), though relatively few studies have been conducted to study CMS-induced neuroadaptations in the mesolimbic dopamine system. The present study reveals that CMS is associated with a decrease in both AP firing and $I_h$ current in VTA dopamine neurons that project to the NAc. Additionally, HCN2 knockdown in the VTA was sufficient to induce depressive- and anxiety-like behavior in non-stressed mice, whereas HCN2 overexpression prevented the CMS-induced development of depressive-like behavior. Thus, HCN2 channels in the VTA play a critical role in regulating depressive- and anxiety-like behavior in both unstressed and chronically stressed mice.

# Materials and methods

### Key resources table

| Reagent type (species) or resource | Designation | Source or reference | Identifiers | Additional information |
|---|---|---|---|---|
| Strain, strain background (*Mus musculus*) | male C57BL/6J mice | The Jackson Laboratory | Stock#: 000664 RRID:IMSR_JAX:000664 | |
| Genetic reagent (*Mus musculus*) | male heterozygous *Slc6a3*$^{Cre+/-}$ (DAT-Cre) mice | The Jackson Laboratory | Stock#: 006660 RRID:IMSR_JAX:006660 | maintained on the C57BL/6J background |
| Genetic reagent (*Mus musculus*) | male Ai9 reporter mice | The Jackson Laboratory | Stock#: 007909 RRID:IMSR_JAX:007909 | maintained on the C57BL/6J background |
| Strain, strain background (*Adeno-associated virus*) | AAV2.shRNA.U6.ShRLuc. CMV.eGFP.SV40 | shRNA provided by Dr. Han-gang Yu (PMID: 19236845), packed into AAV2 at Penn Vector Core | | |
| Strain, strain background (*Adeno-associated virus*) | AAV2.scramble.U6.ShRLuc. CMV.eGFP.SV40 | Penn Vector Core | | |
| Strain, strain background (*Adeno-associated virus*) | AAV2.CMV.PI.HCN2. WPRE.eGFP.SV40 | HCN2 plasmid packed into AAV2 at Penn Vector Core | | HCN2 plasmid provided by Dr. Dane Chetkovich at Northwestern University |

*Continued on next page*

*Continued*

| Reagent type (species) or resource | Designation | Source or reference | Identifiers | Additional information |
|---|---|---|---|---|
| Strain, strain background (*Adeno-associated virus*) | AAV2.CMV.PI.eGFP.WPRE.bGH | Penn Vector Core | | |
| Antibody | Mouse anti-TH | Santa Cruz Biotechnology | SC-136100, Lot: G1309 RRID:AB_2287193 | monoclonal, 1:300 |
| Antibody | Rabbit anti-TH | Santa Cruz Biotechnology | SC-14007, Lot: C2707, RRID:AB_671397 | polyclonal, 1:300 |
| Antibody | Rabbit anti-NeuN | Millipore | ABN78, Lot: 2702139, RRID:AB_10807945 | polyclonal, 1:400 |
| Antibody | anti-mouse IgG Alexa Fluor 555 | Cell Signaling | Stock #4409, RRID:AB_1904022 | Goat anti-mouse, 1:300 |
| Antibody | anti-rabbit IgG Alexa Fluor 647 | Invitrogen | A21245, Lot: 1445259, RRID:AB_141775 | Goat anti-rabbit, 1:100 |
| Antibody | anti-mouse IgG Alexa Fluor 488 | Cell Signaling | Stock #4408, RRID:AB_10694704 | Goat anti-mouse, 1:300 |
| Antibody | anti-rabbit IgG Alexa Fluor 488 | Cell Signaling | Stock #4412, RRID:AB_1904025 | Goat anti-rabbit, 1:500 |
| Other | neurobiotin tracer | Vector Laboratories | SP-1120, RRID:AB_2336606 | 1.5% |
| Other | Texas Red avidin D | Vector Laboratories | A-2006, RRID:AB_2336751 | 1:100 |
| Software (Sigmaplot 11.2) | | | RRID:SCR_003210 | |

## Animals

Animal maintenance and use were in accordance with protocols approved by the Institutional Animal Care and Use Committee of the Medical College of Wisconsin. Mice were given *ad libitum* access to food and water, and housed four to five per cage in a temperature (23 ± 1°C) and humidity-controlled room (40–60%) with a 12 hr light-dark cycle. All experiments were performed on adult male mice (8–10 week-old at the beginning of the experiments). C57BL/6J, heterozygous $Slc6a3^{Cre+/-}$ mice (simplified as DAT-Cre; Jax stock#: 006660), and Ai9 Cre reporter mice (strain code: B6.Cg-Gt (ROSA)26Sor$^{tm9(CAG-tdTomato)Hze}$/J; Jax stock#: 007909,) were obtained from the Jackson Laboratory (Bar Harbor, Maine). All of the mouse lines were maintained on the same C57BL/6J background. By crossing DAT-Cre mice with Ai9 Cre reporter mice, we have generated DAT-tdTomato reporter mice in which tdTomato is selectively expressed in dopamine neurons, which allowed unambiguous identification of tdTomato-fluorescent dopamine neurons during ex vivo slice electrophysiology.

## Animal surgery and microinjection of retrobeads or AAVs

Mice were anesthetized with ketamine (90 mg/kg, i.p.) and xylazine (10 mg/kg, i.p.) and placed in a stereotaxic device (David Kopf Instruments, Tujunga, CA). For retrograde tracing, green Retrobeads (0.1 µl; LumaFluor Inc., Naples, FL) were injected bilaterally in the lateral shell of the NAc of DAT-tdTomato reporter mice (coordinates from bregma: AP 1.45 mm; ML ±1.75 mm; DV −4.4 mm). The following AAVs (0.2 µl) were injected into the VTA bilaterally (coordinates from bregma: AP −3.1 mm; ML ±1.0 mm; DV −4.8 mm at a 7° angle) (*Paxinos and Franklin, 2001*). HCN2-shRNA and scramble-shRNA constructs were provided by Dr. Han-gang Yu at West Virginia University (*Zhang et al., 2009*) and were packaged into AAV2 with an eGFP reporter (AAV2.shRNA.U6. ShRLuc.CMV.eGFP.SV40) at Penn Vector Core, University of Pennsylvania (Philadelphia, PA). The HCN2 plasmid was provided by Dr. Dane Chetkovich at Northwestern University and was packaged into AAV2 at Penn Vector Core (AAV2.CMV.PI.HCN2.WPRE.eGFP.SV40, referred as 'AAV2-HCN-eGFP'). AAV2.CMV.PI.eGFP.WPRE.bGH (referred as 'AAV2-eGFP') serves as a control. The injections of Retrobeads and AAVs were through a Nanoject III Programmable Nanoliter Injector (Drummond Scientific Company, Broomall, PA). The injection rate was 60 nl/min, and the injectors were kept in place for an additional 5 min to ensure adequate diffusion from the injector tip. After the surgery, animals received subcutaneous injections of analgesic (buprenorphine-SR, 1 mg/kg). Mice were

allowed to recover for 1 week before CMS experiments or 3 weeks before slice electrophysiology or immunofluorescence staining unless stated otherwise.

## Chronic unpredictable mild stress (CMS) paradigm

Equal numbers of male C57BL/6J mice and DAT-tdTomato reporter mice (on C57BL/6J background) were subjected to CMS for a total of 5 weeks based on published studies (*Koo and Duman, 2008*; *Willner et al., 1987*). The stressors included restraint (1 hr in a soft, flexible plastic cone, Decapi-Cone, Braintree Scientific, Inc.), inversion of day/night light cycle, cold (in a cold room at 4°C for 1 hr), 45° tilted cage (overnight), cage rotation (20 min), rat bedding (odor, 3 hr), wet bedding (250 ml water added into cage, overnight), no bedding (overnight), low intensity stroboscopic illumination (10 Hz, overnight), food and water deprivation (overnight), and overcrowding (overnight). Two stressors were administered per day. The timeline of the stressor exposure has been described in detail in our recent studies (*Zhong et al., 2014a*, *2014b*) and also in *Table 1*. Non-stressed controls were handled only for cage changes and behavioral tests.

## Behavior

Behavioral tests have been described in detail in our recent studies (*Zhong et al., 2014a*, *2014b*). Behavioral testing was carried out by experimenters blind to genotype, group, and/or treatment history, and less stressful behavioral tests were typically tested before more stressful behavioral tests. Behavioral tests were conducted in the order listed below, with only one behavioral test conducted per day.

### Open field test (OFT)

Mice were placed individually in one corner of an open field (50 cm length x 45 cm wide x 30 cm deep box) and allowed to freely explore the arena during a 20 min test session. Locomotor activity was recorded using an automated video-tracking system (Mobile Datum, Shanghai, China). Total distance traveled and time spent in the center of the box during the first 5 min was calculated. Center time is defined as the amount of time that was spent in the central 25 cm x 22.5 cm area of the open field.

### Sucrose preference test (SPT)

Mice were individually housed and trained to drink from two drinking bottles for 48 hr. One bottle contained 1% sucrose (in tap water) and the other contained tap water. The SPT was carried out after the OFT. During the SPT, mice were deprived of food and water for 8 hr, and the consumption of sucrose solution and water over the next 16 hr was measured. Sucrose preference (%) was calculated as sucrose solution consumed divided by the total amount of solution consumed.

**Table 1.** Experimental schedule for the chronic mild stress (CMS) procedure in mice

| Week | Monday | Tuesday | Wednesday | Thursday | Friday | Saturday | Sunday |
|---|---|---|---|---|---|---|---|
| 1 | Cold Wet bedding | Restraint No bedding | Light inversion Cage tilt | Cage rotation Strobe | Cold Food and water deprivation | Restraint Overcrowding | Light inversion Wet bedding |
| 2 | Cold Cage tilt | Cage rotation Food and water deprivation | Restraint Wet bedding | Rat bedding Strobe | Light inversion No bedding | Cage rotation Food and water deprivation | Cold Wet bedding |
| 3 | Rat bedding Strobe | Restraint Light inversion | Cage rotation No bedding | Light inversion Food and water deprivation | Cold Wet bedding | Cage tilt Strobe | Light inversion Overcrowding |
| 4 | Cold No bedding | Restraint Food and water deprivation | Cage rotation Strobe | Rat bedding Light inversion | Cold Cage tilt | Restraint Wet bedding | Cage rotation No bedding |
| 5 | Cold Food and water deprivation | Cage rotation Strobe | Light inversion Wet bedding | Cold Cage tilt | Cage rotation No bedding | Light inversion Overcrowding | Restraint Cage tilt |

DOI: https://doi.org/10.7554/eLife.32420.020

## Elevated plus maze (EPM)

The EPM apparatus (Stoelting, Wood Dale, IL) consists of two open arms (35 × 5 cm) across from each other and perpendicular to two closed arms (35 × 5 × 15 cm) that are connected by a center platform (5 × 5 cm). The apparatus is elevated 40 cm above the floor. Mice were placed in the center platform facing a closed arm and allowed to freely explore the maze for 5 min. The location of the mice was tracked with the automated video-tracking system. The percent of entries into open arms and time spent in open arms were analyzed.

## Novelty-suppressed feeding (NSF)

The NSF test was carried out similar to a published protocol (*Santarelli et al., 2003*). Mice were food deprived for 24 hr before being placed in a novel environment (a plastic box 45 cm long x 35 cm wide x 20 cm deep) where five food pellets (regular chow) were placed on a piece of white filter paper (11 cm in diameter) in the center of the box. A mouse was placed in one corner of the box and the latency to feed was measured. Feeding was defined as biting the food with the use of forepaws, but not simply sniffing or touching the food. Immediately after the test, the animal was transferred to the home cage, and the latency to feed in the home cage was measured to serve as a control.

## Forced swim test (FST)

Mice were placed individually into glass cylinders (13 cm diameter, 25 cm tall) filled to a depth of 18 cm with water (30 ± 1°C). The mice were placed in the cylinders for 6 min. The time spent immobile during the last 4 min was scored. Immobility was defined as the cessation of all movements (e.g., climbing, swimming) except those necessary for the mouse to keep its head above water (i.e., floating).

## In vivo electrophysiology

One day after the last behavioral test, CMS and time-matched control mice were anesthetized with urethane (1.5 mg/kg, i.p.) and were positioned in a stereotaxic frame (David Kopf Instruments). Their body temperature was maintained at 37°C using a heating pad. Craniotomies were performed to allow single-unit recordings of VTA dopamine neurons. The areas for electrode insertion were moisturized with saline. Single unit recording electrodes were pulled from micropipettes (O.D., 1 mm, I.D., 0.5 mm) to a resistance of 10–15 MΩ when filled with 2 M NaCl containing 1.5% neurobiotin. The electrode was lowered into the VTA (coordinates from bregma: AP −2.9 to −3.3 mm, ML 0.6 to 1.1 mm, DV −3.9 to −4.5 mm) through a micromanipulator, and a reference electrode was placed in the subcutaneous tissue. These coordinates primarily correspond to the lateral parabrachial pigmented nucleus (PBP) of the VTA. Single-unit activity was acquired with a Multiclamp 700B amplifier and a DigiData 1440A digitizer and was analyzed by pClamp 10.3 (Molecular Devices). Signals were sampled at 10 kHz, and the bandpass filter was set between 0.3 and 5 kHz (*Bocklisch et al., 2013*; *Brischoux et al., 2009*; *Chaudhury et al., 2013*). Dopamine neurons were identified by a broad triphasic extracellular action potential of a width greater than 2 ms and a relatively slow firing rate (<10 Hz) (*Ungless et al., 2004*). Burst firing was defined as beginning when two action potentials have an inter-spike interval of <80 ms and ending when two action potentials have an inter-spike interval of >160 ms (*Bishop et al., 2010*; *Chen and Lodge, 2013*; *Grace and Bunney, 1984*; *Schiemann et al., 2012*). These criteria provide reliable identification of dopamine neurons in vivo, as determined by juxtacellular labeling and colocalization with tyrosine hydroxylase (TH) (*Ungless and Grace, 2012*; *Ungless et al., 2004*). Nevertheless, we did juxtacellular labeling with neurobiotin to verify the identity of dopamine neurons. Briefly, following the last electrophysiological recording in each mouse, positive current pulses (7 s on/off cycles) were applied through the recording electrode to the neuron for 4–10 min. The neurobitotin was allowed to transport within the neuron for another 1–2 hr before the animals were prepared for immunofluorescence staining (*Bocklisch et al., 2013*; *Pinault, 1996*). To avoid potential effects of the labeling procedure on neuron firing, no additional neurons were recorded following neurobiotin electroporation.

## Slice preparation and electrophysiology

One day after the last behavioral test, CMS and time-matched control DAT-tdTomato reporter mice were anaesthetized by isoflurane inhalation and decapitated. The brain was trimmed and embedded in 3% low-melting-point agarose, and horizontal midbrain slices (200 µm thick) were cut using a vibrating slicer (Leica VT1200s, Nussloch, Germany), as described in our recent studies (*Liu et al., 2016*; *Tong et al., 2017*). Slices were prepared in a choline-based solution containing (in mM): 110 choline chloride, 2.5 KCl, 1.25 $NaH_2PO_4$, 0.5 $CaCl_2$, 7 $MgSO_4$, 26 $NaHCO_3$, 25 glucose, 11.6 sodium ascorbate, and 3.1 sodium pyruvate at room temperature. The VTA slices were cut in the midline to produce two individual slices from each section. The slices were incubated for 30 min in sucrose-based solution containing (in mM): 78 NaCl, 68 sucrose, 26 $NaHCO_3$, 2.5 KCl, 1.25 $NaH_2PO_4$, 2 $CaCl_2$, 2 $MgCl_2$ and 25 glucose. Then, the slices were allowed to recover for at least 30 min in the artificial cerebrospinal fluid (ACSF) containing (in mM): 119 NaCl, 2.5 KCl, 2.5 $CaCl_2$, 1 $MgCl_2$, 1.25 $NaH_2PO_4$, 26 $NaHCO_3$, and 10 glucose. All solutions were saturated with 95% $O_2$ and 5% $CO_2$.

Whole-cell and cell-attached patch-clamp recordings were made as described in our previous study (*Zhong et al., 2015*). Recordings were made using patch-clamp amplifiers (Multiclamp 700B) under infrared differential interference contrast (DIC) microscopy. Data acquisition and analysis were performed using DigiData 1440A and 1550B digitizers and the analysis software pClamp 10.3 (Molecular Devices). Signals were filtered at 2 kHz and sampled at 10 kHz. $I_h$ current was measured by inducing 1.5 s hyperpolarizing steps from −60 mV to −130 mV with −10 mV steps. Junction potentials between the patch pipette and bath ACSF were nullified prior to obtaining a seal. For the generation of $I_h$ activation curves, 1.5 s hyperpolarizing steps to various potentials (−60 to −130 mV) were applied from a holding potential of −60 mV and tail currents were measured at −130 mV. Tetraethylammonium chloride (TEA-Cl, 10 mM) was included in the ACSF to block non-inactivating voltage-dependent $K^+$ conductance and osmolality was maintained by equimolar reduction of NaCl from the ACSF. Tail current amplitudes at −130 mV, after subtraction of the current following no hyperpolarizing step, were plotted as a function of test potentials. The $I_h$ activation curve was fitted with a Boltzmann function $I = I_{max}/\exp[(V_m – V_{1/2})/s]$, where $I_{max}$ is the maximal tail current amplitude, $V$ is the test potential, $V_{1/2}$ is the half-activation potential, and $s$ is the slope factor. Resting membrane conductance was approximated from voltage clamp recordings in a manner similar to (*Kimura et al., 1988*). Hyperpolarizing voltage steps from a resting holding potential of −60 mV to −130 mV in 10 mV steps generated instantaneous inward currents ($I_{ins}$). $I_{ins}$ was plotted against the hyperpolarizing voltage steps. The slope of these I-V curves provides an approximation of the resting membrane conductance ($G_{resting}$). Membrane capacitance was measured by Clampex software (Molecular Devices) using small amplitude hyperpolarizing and depolarizing steps (±5 mV). Firing rate was recorded in the cell-attached configuration in the presence of CNQX (10 µM), D-AP5 (20 µM) and picrotoxin (50 µM) to block excitatory and inhibitory synaptic transmission. Glass pipettes (3–5 MΩ) were filled with an internal solution containing (in mM): 140 K-gluconate, 10 KCl, 10 HEPES, 0.2 EGTA, 2 $MgCl_2$, 4 Mg-ATP, 0.3 $Na_2$GTP (pH 7.2 with KOH). Series resistance (10–20 MΩ) was monitored throughout all recordings, and data were discarded if the resistance changed by more than 20%. All recordings were performed at 32 ± 1°C by using an automatic temperature controller (Warner Instruments, Inc.).

## Immunofluorescence staining

Mice were anaesthetized by ketamine (90 mg/kg, i.p.) and xylazine (10 mg/kg, i.p.) and transcardially perfused with 0.1 M sodium phosphate buffered saline (PBS) followed by 4% paraformaldehyde in 4% sucrose-PBS (pH 7.4). After perfusion, the brain was removed and post-fixed in the same fixative for 4 hr at 4°C, and was then dehydrated in increasing concentrations of sucrose (20% and 30%) in 0.1 M PBS at 4°C and frozen on dry ice. Coronal VTA sections (20 µm) were cut with a Leica cryostat. VTA sections were incubated with antibodies against tyrosine hydroxylase (TH, mouse, 1:300, Santa Cruz Biotechnology) and/or NeuN (rabbit, 1:400; Millipore) at 4°C for 48 hr. VTA sections were then incubated with anti-mouse IgG Alexa Fluor 555-conjugated (goat, 1:300; Cell Signaling) and anti-rabbit IgG Alexa Fluor 647-conjugated (goat, 1:100; Invitrogen) or anti-mouse IgG Alexa Fluor 488-conjugated (goat, 1:300; Cell Signaling) secondary antibodies for 4 hr at room temperature in the dark. To reveal labeled cells using the juxtacellular method, VTA sections were first incubated with Texas Red avidin D (1:100; Vector Laboratories) to retrieve the labeled cell bodies. Then, the selected

tissue sections were incubated with anti-TH (rabbit, 1:300; Santa Cruz Biotechnology; 48 hr) and anti-rabbit IgG Alexa Fluor 488-conjugated (goat, 1:500; Cell Signaling; 4 hr). Confocal imaging was performed using a Nikon TE2000-U inverted microscope equipped with the C1 Plus confocal system.

## Chemicals

Picrotoxin, TTX and all other common chemicals were obtained from Sigma-Aldrich (St. Louis, MO). 6-cyano-7-nitroquinoxaline-2,3-dione disodium salt (CNQX), D-(-)-2-Amino-5-phosphonopentanoic acid (D-AP5) and ZD7288 were obtained from Tocris Bioscience (Ellisville, MO).

## Data analysis and statistics

All results are presented as the mean ±SEM. Results were analyzed with either Student's *t*-test, or two-way ANOVA followed by Tukey's *post hoc* analysis using Sigmaplot 11.2. Results were considered to be significant at p<0.05.

## Acknowledgements

This work was supported by NIH Grants DA035217 and MH101146 to Q.s.L, and by F30MH115536 to CRV. It was also partially funded through the Research and Education Initiative Fund, a component of the Advancing a Healthier Wisconsin endowment at the Medical College of Wisconsin. CRV is a member of the Medical Scientist Training Program at MCW, which is partially supported by a training grant from NIGMS T32-GM080202.

## Additional information

### Funding

| Funder | Grant reference number | Author |
|---|---|---|
| National Institute of Mental Health | F30MH115536 | Casey R Vickstrom |
| National Institute on Drug Abuse | DA035217 | Qing-song Liu |
| National Institute of Mental Health | MH101146 | Qing-song Liu |
| Medical College of Wisconsin | Research and Education Initiative Fund, Advancing a Healthier Wisconsin (AWH) | Qing-song Liu |

The funders had no role in study design, data collection and interpretation, or the decision to submit the work for publication.

### Author contributions

Peng Zhong, Conceptualization, Data curation, Formal analysis, Validation, Investigation, Methodology, Writing—original draft, Writing—review and editing; Casey R Vickstrom, Conceptualization, Data curation, Funding acquisition, Validation, Investigation, Methodology, Writing—original draft, Writing—review and editing; Xiaojie Liu, Conceptualization, Data curation, Formal analysis, Investigation, Methodology, Writing—original draft, Writing—review and editing; Ying Hu, Data curation, Formal analysis, Investigation, Methodology; Laikang Yu, Data curation, Formal analysis, Investigation, Writing—original draft; Han-Gang Yu, Conceptualization, Resources, Validation, Visualization; Qing-song Liu, Conceptualization, Resources, Supervision, Funding acquisition, Investigation, Project administration, Writing—review and editing

### Author ORCIDs

Casey R Vickstrom (ID) http://orcid.org/0000-0003-0536-1211
Han-Gang Yu (ID) http://orcid.org/0000-0001-6838-8310
Qing-song Liu (ID) http://orcid.org/0000-0003-1858-1504

### Ethics

Animal experimentation: Animal maintenance and use were in accordance with protocols approved by the Institutional Animal Care and Use Committee of the Medical College of Wisconsin (#1166, #2420).

### Decision letter and Author response

Decision letter https://doi.org/10.7554/eLife.32420.024
Author response https://doi.org/10.7554/eLife.32420.025

## Additional files

### Supplementary files

• Supplementary file 1. Statistical results for *Figure 1—figure supplement 1*. Abbreviations: CMS, chronic mild unpredictable stress; EPM, elevated plus maze; FST, forced swimming test; NSF, novelty-suppressed feeding; OFT, open field test; SPT, sucrose preference test
DOI: https://doi.org/10.7554/eLife.32420.021

• Transparent reporting form
DOI: https://doi.org/10.7554/eLife.32420.022

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
