## [Decision Letter]

Thank you for submitting your article "HCN2 Channels in the Ventral Tegmental Area Regulate Behavioral Responses to Chronic Stress" for consideration by *eLife*. Your article has been reviewed by three peer reviewers, and the evaluation has been overseen by Olivier Manzoni as the Reviewing Editor and Gary Westbrook as the Senior Editor. The following individuals involved in review of your submission have agreed to reveal their identity: John T Williams (Reviewer #2); Miriam Melis (Reviewer #3).

The reviewers have discussed the reviews with one another and the Reviewing Editor has drafted this decision to help you prepare a revised submission. We hope you will be able to submit the revised version within two months.

Summary:

Zhong et al. investigated the synaptic and molecular mechanisms underlying chronic mild stress (CMS)-induced reduction in Ventral Tegmental Area (VTA) dopamine (DA) neuron firing.

The authors used a combination of behavior, electrophysiology, and RNA knock-down/overexpression to investigate changes in overexpressed hyperpolarization – activated cyclic nucleotide – gated channel (HCN) currents after CMS, and the effect of manipulating HCN2 channels on behavior. The data first reveal that CMS causes anxiety- or depressive-like behavior, which is associated with reduced DA neuron activity in the VTA. Second, the results show that CMS reduces synaptic excitation of VTA DA neurons and HCN current. Finally, knock-down of HCN2 in the VTA was sufficient to induce depressive-like behavior in mice and over-expression of HCN2 in the VTA was sufficient to prevent depressive-like behavior in response to CMS.

Essential revisions:

1) In their current state, the experiments studying synaptic parameters (i.e. minis, AMPA/NMDA ratio…) are: (1) per se not sufficient to provide mechanistic insights (2) rather confusing (see reviewers' comments) (3) not sufficiently discussed. Thus, we advise that you remove this data set (and the corresponding discussion) from your revision. Alternatively, although it is not the reviewer's preferred option, the authors may choose performing all additional experiments suggested by the reviewers in their detailed reviews.

Also, for the sake of clarity, the Results section measuring the sag of membrane potential are really not necessary and should be removed.

2) The VTA contains a heterogeneous population of DA neurons with respect to their projection targets and HCN channel expression. Specifically, neurons that project to the nucleus accumbens exhibit large HCN currents, whereas those that project to the prefrontal cortex do not.

It is imperative that the authors analyze (Results section) and interpret (Discussion section) their results in light of the different target populations and different cell types that were identified in the VTA. Notably, the paper by Friedman et al., 2014 where Ih and VTA neuronal excitability were studied in another stress paradigm and the article by Moreines et al., (2017) where reduced activity was observed in a subpopulation of VTA DA cells in a CMS model, must be discussed. Additionally, because of the "laterality" issue raised by Moreines et al., the authors are requested to document and discuss the location of the recorded DA cells (both ex vivo and in vivo).

3) The revised version must include the following technical details:

-The resting conductance of the neurons must be provided in a table. The conductance of the cells illustrated in Figure 4 and Figure 6 are very different.

-The capacitance of the dopamine cells (50 pS) is about twice that which is often reported. How was the capacitance of the cells measured? What were the setting for the acquisition and filtering frequencies?

-In the figure legends, the statements of the number of n per group are not clear. For example, in Figure 2, it states "n=10-13 neurons from 4-5 mice". Does this mean 4-5 mice per group (control vs CMS), or combined between both groups?

-The authors state that the C57BL/6 mice and the DAT-tdTomato mice did not show significant differences in their behavioral tests, so the results were pooled. The authors should show supplementary data to demonstrate this claim.

Reviewer #1:

In this paper, Zhong et al. investigate how HCN2 channels in the Ventral Tegmental Area (VTA) regulate depressive-like behavior in response to chronic mild stress (CMS). The authors use a combination of behavior, electrophysiology, and RNA knock-down/overexpression to investigate changes in HCN current after CMS, and the effect of manipulating HCN channels on animals' behavior. The authors present a number of important findings. First, they report that CMS causes anxiety- or depressive-like behavior, which is associated with reduced DA neuron activity in the VTA. Then, they report that CMS also causes reduced synaptic excitation of VTA DA neurons, as well as reduced HCN current. They went on to preform behavioral experiments to demonstrate that knock-down of HCN2 in the VTA is sufficient to induce depressive-like behavior in mice and that over-expression of HCN2 in the VTA is sufficient to prevent depressive-like behavior in response to CMS. This is an interesting paper with several important findings, but there are several unclear parts at this point

One unclear part with the study is a lack of mechanistic understanding of how modulation of HCN channels regulates DA neuron activity, in particular regarding the relationship between the observed changes in synaptic and intrinsic excitability. The authors show effects in minis and AMPA/NMDA ratio, but these findings are not followed up and are not put in the context of HCN channels. HCN channels can have effects on synaptic and intrinsic excitability (input resistance, resting membrane potential, etc.) but this is neither addressed or discussed

As a result, as it is currently presented, the synaptic excitation data is out of place and it is confusing how it is related, if at all, to the main finding regarding changes in HCN channels. There is only a very brief discussion mentioning that changes in synaptic input, as well as changes in HCN, may work together to underlie behavioral output.

The writing is also confusing: are the authors after the intrinsic and synaptic properties mediating the changes in the excitability of VTA neurons or are the after the role of H channels. The Intro was confusing in this regard too, as it does not prepare the reader as to what to expect

One other unclear point is whether the authors intend to focus on VTA DA neuron activity as a whole population, or specifically the population of neurons that project to the nucleus accumbens (NAc). Because the experiments here target various neuron populations in the VTA, this complicates the understanding and the interpretation of results in the context of a heterogeneous population of VTA DA neurons.

The VTA contains a heterogeneous population of DA neurons with respect their projection targets and HCN channel expression. Specifically, neurons that project to the nucleus accumbens (NAc) exhibit large HCN currents, whereas those that project to the prefrontal cortex do not. However, in Figure 2, Figure 3, and 8, the authors seemingly record from the entire VTA population, while in Figure 4, Figure 5, and 6, they specifically target the NAc-projecting population, which exhibit large HCN currents. (Note: in the Figure 3 legend, it states that the recordings are from NAc-projecting neurons, but this is not discussed in the text describing these experiments.) Therefore, the rationale for this experimental design is unclear.

This confusion does not allow for a clear understanding and assessment of the novelty of this work, especially in relationship to the paper by Friedman et al., 2014, which also deals with Ih and VTA neuronal excitability in another stress paradigm. This paper shows several different and similar results to this paper. Although the authors mention this paper several times, the rationale for their experimental design (cell types etc.) and the discussion of their results is not clear; therefore, I was unable to form a clear picture on the role of VTA HCN channels in behavioral responses to chronic stress.

Reviewer #2:

This manuscript examines the regulation of dopamine neurons that project to a specific area of the nucleus accumbens in control and following treatment of animals with manipulations that induce chronic stress. The results show that the animals show signs of stress induce depression, the activity of dopamine neurons is decreased and there is a reduction in I-h. Decreasing I-h with sh-RNA results in depressive like behaviors and expressing I-h prevents the depressive like behaviors.

1) This is careful work looking at dopamine neurons using appropriate models and the inclusion of projection specific neurons.

2) There are a few details that should be included in the manuscript. One easy and important addition is the inclusion of the resting conductance of the neurons. Those results are surely available. The reason for this request is based on the figures illustrating the differences in I-h. The conductance of the cells illustrated in Figure 4 are very different. Likewise there is a huge difference in the conductance of the cells illustrated in Figure 6. These may just be bad examples but having a table of the conductance’s would be very helpful.

3)The capacitance of the dopamine cells (50 pS) is about twice that which is often reported. How was the capacitance of the cells measured? What were the setting for the acquisition and filtering frequencies?

4) The Results section measuring the sag of membrane potential are really not necessary and should be removed.

5) There is a considerable amount of discussion in the results that is repeated in the discussion. The Results section could be edited to remove most of that and the manuscript would be more readable.

Reviewer #3:

In this study Zhong and colleagues examine the biophysical mechanisms underlying CMS-induced depressive phenotype in mice. This is a potentially important study. Hence, it provides the first clear demonstration of a critical role of HCN2 in dynamically shaping NAc-projecting VTA DA AP firing activity in CMS mouse model by combining in vivo and ex vivo electrophysiology, and virus-mediated knockdown/overexpression of HCN2 in the lateral VTA. The manuscript is well-organized, clearly and thoroughly written, and the level of technical detail is appropriate.

I have a couple of concerns and suggestions listed below.

First, based on their findings the authors suggest that a reduced excitatory tone might contribute to the decreased DA AP firing in vivo. However, a reduction in both frequency and amplitude of mEPSCs together with a decreased AMPA/NMDA ratio is not as convincing as they state, since it is suggestive of other postsynaptic mechanisms to be involved rather than HCN2 solely. A paired pulse protocol and a measure of CV might add potential insights for the identification of this mouse phenotype.

Second, since the Authors found a reduced population activity they should cite the article by Moreines et al., (2017) where the same effect is observed in a CMS model but only in a subpopulation of VTA DA cells. Additionally, because of the "laterality" issue raised by Moreines et al. they should also discuss the location of their recorded DA cells (both ex vivo and in vivo). Finally, they could add a figure (or a table/panel) with their recording sites. This will add critical value to the study especially in light of the heterogeneity of VTA DA cells. Specie-specific effects might be also taken into account and discussed.

---

## [Author Response]

Essential revisions:1) In their current state, the experiments studying synaptic parameters (i.e. minis, AMPA/NMDA ratio…) are: (1) per se not sufficient to provide mechanistic insights (2) rather confusing (see reviewers' comments) (3) not sufficiently discussed. Thus, we advise that you remove this data set (and the corresponding discussion) from your revision. Alternatively, although it is not the reviewer's preferred option, the authors may choose performing all additional experiments suggested by the reviewers in their detailed reviews.Also, for the sake of clarity, the Results section measuring the sag of membrane potential are really not necessary and should be removed.

We have removed the experiments studying synaptic parameters (original Figure 3), membrane potential sag (original Figure 4), and the corresponding discussions.

2) The VTA contains a heterogeneous population of DA neurons with respect to their projection targets and HCN channel expression. Specifically, neurons that project to the nucleus accumbens exhibit large HCN currents, whereas those that project to the prefrontal cortex do not.

*It is imperative that the authors analyze (Results section) and interpret (Discussion section) their results in light of the different target populations and different cell types that were identified in the VTA. Notably, the paper by Friedman et al., 2014 where Ih and VTA neuronal excitability were studied in another stress paradigm and the article by Moreines et al., (2017) where reduced activity was observed in a subpopulation of VTA DA cells in a CMS model, must be discussed. Additionally, because of the "laterality" issue raised by Moreines et al., the authors are requested to document and discuss the location of the recorded DA cells (both* ex vivo *and* in vivo).

The reviewer is correct that the VTA contains a heterogeneous population of dopamine neurons in regard to many aspects, including projection targets and HCN expression. Given that HCN current is the focus of our present study, our in vivo and ex vivo recordings targeted the lateral parabrachial pigmented area (PBP) of the VTA, where dopamine neurons exhibit large HCN currents, predominantly project to the lateral shell of the NAc and play a primary role in reward and motivated behavior. Our ex vivo recordings were made from NAc lateral shell-projecting DA neurons labelled with retrobeads, which were predominantly located in the lateral PBP of the VTA, consistent with Lammel et al., (2008, 2011). We also made efforts to target DA neurons in the lateral PBP in our in vivo recordings. The coordinates used for in vivo recordings (AP -2.9 to -3.3 mm, ML 0.6 to 1.1 mm, DV -3.9 to -4.5 mm) correspond to the PBP (mainly lateral PBP) but not in midline nuclei such as the interfascicular nucleus and the rostral linear nucleus, nor A10 DA neurons in the supramammillary nucleus. The location of some neurons was confirmed by juxtacellular labeling with neurobiotin and TH staining, with one neuron labeled in each mouse. To clarify the location of recorded DA neurons, we have revised the manuscript to more explicitly state that recordings were targeted to the lateral PBP.

Additionally, we provide a clearer rationale for targeting the lateral PBP and NAc lateral shell-projecting DA neurons, and better discuss our results in the context of VTA DA neuron heterogeneity. We specifically addressed the “laterality” issue raised in the Moreines et al. study. We suspect that differences in CMS paradigms and animal species might explain why Moreines et al. did not observe alterations in lateral VTA dopamine neuron firing. In addition, we have further discussed our results in the context of Friedman et al., 2014. In particular, we discuss effects on I_h_ current in DA neurons that project to different targets, and discuss the circuit-specific effects on HCN overexpression on depressive-like behavior. Further, we make clear the limitations of our viral knockdown and overexpression approaches, and discuss opportunities for future investigation into cell- and circuit-specific contributions of HCN2 channels in depressive- and anxiety-like behavior.

3) The revised version must include the following technical details:-The resting conductance of the neurons must be provided in a table. The conductance of the cells illustrated in Figure 4 and Figure 6 are very different.

We analyzed the instantaneous current (I_ins_) in traces of I_h_ current in Figure 4 and Figure 6, and plotted this current against the hyperpolarizing voltage steps. The slope of these I-V curves provides an approximation of the resting membrane conductance (G_resting_) (the measurement of G_resting_ has been shown in Figure 4—figure supplement 1). We find that there was not a significant effect of CMS on resting membrane conductance compared to control (Figure 4). However, HCN2 knockdown with shRNA reduced resting membrane conductance compared to scramble-shRNA (Figure 6—figure supplement 1). We have replaced traces for I_h_ current in Figure 4 and Figure 6 that better represent the averaged data. We have described the design and interpretation of the resting membrane conductance in the Materials and methods, Results and Discussion sections.

-The capacitance of the dopamine cells (50 pS) is about twice that which is often reported. How was the capacitance of the cells measured? What were the setting for the acquisition and filtering frequencies?

VTA dopamine neurons have a relatively large cell body compared with many other cell types. We performed a literature search to determine what previous studies have measured for the capacitance of VTA dopamine neurons in mice. We find that the capacitance we recorded (control, ~51 pF; CMS, ~45 pF) is consistent with published studies (Chung et al., 2017, PMID: 28894175; Baimel et al., 2017, PMID: 28178514; Zhang et al., 2010, PMID: 20600174). Chung et al., (2017) reported the capacitance for VTA dopamine neurons to be ~50-60 pF. Baimel et al., (2017) reported differences in VTA dopamine neuron capacitance for different projection-defined dopamine neurons. Consistent with our result, the average capacitance for NAc lateral shell-projecting DA neurons was 57 pF. Although other populations had lower capacitances (24 pF for NAc medial shell-projecting; 38 pF for BLA-projecting), our recordings focused on NAc lateral shell-projecting neurons. In a third study, Zhang et al., (2010) report a capacitance of 79 pF for VTA DA neurons in the lateral VTA and 54 pF in the medial VTA. Thus, our result does not differ substantially from published studies. Membrane capacitance was measured by Clampex software using small amplitude hyperpolarizing and depolarizing steps ( ± 5 mV). As mentioned in the Materials and methods section, signals were sampled at 10 kHz and filtered at 2 kHz.

-In the figure legends, the statements of the number of n per group are not clear. For example, in Figure 2, it states "n=10-13 neurons from 4-5 mice". Does this mean 4-5 mice per group (control vs CMS), or combined between both groups?

In each relevant figure legend, we have clarified the numbers of neurons and mice in each group to avoid confusion.

-The authors state that the C57BL/6 mice and the DAT-tdTomato mice did not show significant differences in their behavioral tests, so the results were pooled. The authors should show supplementary data to demonstrate this claim.

We now provide Figure 1—figure supplement 1 presenting this data and Figure 1—source data 1 for statistical analysis of this data.